# The effect of mountain uplift on eastern boundary currents and upwelling systems

Gerlinde Jung[1,2], Matthias Prange[1,2]

[1]MARUM - Center for Marine Environmental Sciences, University of Bremen, 28334 Bremen, Germany
[2] Faculty of Geosciences, University of Bremen, 28334 Bremen, Germany

*Correspondence to*: Gerlinde Jung (gjung@marum.de)

**Abstract.** All major mountain ranges are assumed to have been subject to increased uplifting processes during the late Miocene and Pliocene. Previous work has demonstrated that African uplift is an important element to explain Benguela upper-ocean cooling in the late Miocene/Pliocene. According to proxy records, a surface ocean cooling also occurred in
other Eastern Boundary upwelling regions during the late Neogene. Here we investigate a set of sensitivity experiments altering topography in major mountain regions (Andes, North American Cordillera and South/East African mountains) separately with regard to the potential impact on the intensity of near-coastal low-level winds, Ekman transport and Ekman pumping as well as upper-ocean cooling. The simulations show that mountain uplift is important for upper-ocean temperature evolution in the area of Eastern Boundary Currents. The impact is primarily on the atmospheric circulation
which is then acting on upper-ocean temperatures through changes in strengths of upwelling, horizontal heat advection and surface heat fluxes. Different atmosphere-ocean feedbacks additionally alter the sea surface temperature response to uplift. The relative importance of the different feedback mechanisms depends on the region, but is most likely also influenced by model and model resolution.

## 1 Introduction

Eastern Boundary Upwelling Systems (EBUSs) are found off the West coasts of all continents bordering Atlantic and Pacific Ocean. These upwelling systems are characterized by high biological activity and the world's richest fishing grounds due to the upwelling of cool subsurface nutrient-rich waters (Garcia-Reyes and Largier, 2012). The upwelling systems also influence climatic patterns of the land region because local sea surface temperature (SST) is strongly linked to rainfall distribution (e.g. Rouault et al., 2003).

Coastal upwelling is mainly occurring as a response to Ekman transport by alongshore winds and due to wind-stress-curl-driven ocean-surface divergence offshore (in this study we refer to both processes as Ekman pumping and define upward pumping, sometimes called "Ekman suction", to be positive). Strength of upwelling also depends on coastal mountain topography that has an influence on the intensity of coastal low-level jets (Ma et al., 2019). Furthermore, a strong feedback exists between near-coastal winds and SST (Nicholson, 2010;Garcia-Reyes and Largier, 2012), since SST has a strong

influence on the wind field through atmospheric boundary layer stability and dynamics. The major Eastern Boundary Currents are formed by equatorward directed winds at the eastern flanks of the Subtropical Gyres. Due to the presence of the coastline the associated winds induce an Ekman-driven upwelling near the coast (Chavez and Messie, 2009). Apart from coastal and large-scale winds, also offshore and remote ocean conditions (e.g. the strength and depth of the thermocline) influence upwelling activity(Garcia-Reyes and Largier, 2012). The characteristics of EBUSs are also determined by water column stratification, coastal topography and latitudinal position (i.e. influencing seasonality and through the latitudinal dependence of the Coriolis parameter)(Chavez and Messie, 2009). Additionally, the size of the ocean basin is important, where smaller basins, like the Atlantic are less influenced by thermocline perturbations than the larger Pacific Ocean (Chavez and Messie, 2009).

Proxy data from all major EBUSs of the world indicate a strengthening of upwelling intensity during the late Neogene (Dekens et al., 2007). The South American (Strecker et al., 2007), as well as the North American and African continents were situated at approximately the same geographic location throughout most of the Neogene (Scotese, 1998), hence an upwelling intensification through continental shifts is not a very likely scenario. Different proxy studies indicate that all major near-coastal mountain ranges showed significant uplift during the period of upwelling intensification (see below). As mountains act in enhancing the land-ocean temperature gradient, mountain uplift has an impact on thermal wind and baroclinity in the atmospheric boundary layer. Therefore, topographic uplift of, especially, the coastal regions might well serve to intensify upwelling in EBUSs, as was demonstrated by Jung et al. (2014) for the Benguela upwelling system.

In the present study we use a coupled ocean-atmosphere general circulation model GCM) including a dynamic vegetation module. Different model experiments were performed altering topography of East and Southern Africa, the Andes, and the North American Cordillera separately. In these sensitivity experiments topography is the only boundary condition that is altered. Therefore this study aims at understanding the role of continental-scale topography uplift in Neogene upwelling intensification in three major Eastern Boundary upwelling systems. Since these experiments are sensitivity experiments, altering one boundary condition at a time, the major goal is to gain process understanding and we do not aim at representing a specific time in Earth's history. We compare the responses of 1) the Humboldt Current and Upwelling System to Andean uplift, 2) the California Upwelling System to an uplift of the North American Cordillera, and 3) the Benguela Upwelling System to African uplift (as discussed in Jung et al. (2014)).

## 2 Eastern Boundary Upwelling Systems

## 2.1 Modern Eastern Boundary Upwelling Systems

The Benguela Upwelling System is confined by the Angola-Benguela-Front to the North and by the Agulhas Current to the South. The upwelling cell at Lüderitz persists all year, whereas North of Lüderitz the main upwelling season is in austral winter and South of Lüderitz in austral summer (Hutchings et al., 2009). The main zone of year-round upwelling reaches

from ~15°S to ~30°S (Chavez and Messie, 2009), and the most important driver of Benguela upwelling is the wind stress of the low-level Benguela Jet (Nicholson, 2010).

The California upwelling region contains several upwelling regimes and extends from the North Pacific current to off Mexico. The zone of strongest upwelling lies between ~34°N and ~44°N (Chavez and Messie, 2009). It is forced by the North Pacific High, which is especially strong during boreal summer season. California upwelling is characterized by a strong seasonality with the main upwelling season occurring from April to June with strong upwelling favourable winds of the low-level jet (Garcia-Reyes and Largier, 2012). The formation and dynamics of the coastal jet along the California coast has extensively been studied (e.g. Holt, 1996;Doyle, 1997;Burk and Thompson, 1996;Parish, 2000). In winter the Aleutian Low is dominantly strong, leading to winds directed to the North and hence favouring downwelling in most of the area. Only South of about Cape Mendicino there is a year-round, but weak upwelling zone (Checkley and Barth, 2009).

The EBUS of the South Pacific Ocean in the region of the Humboldt Current (also called the Peru Current) is referred to as the Humboldt (or Peru-Chile) Upwelling System. Coastal upwelling is quasi-permanent off Peru and northern Chile, but shows a strong seasonality south of 30°S with maximum upwelling-favourable winds in austral summer (Montecino and Lange, 2009). Only Peru upwelling shows a year-round activity between ~4°S to ~16°S (Chavez and Messie, 2009), but it is periodically disrupted by the El Nino Southern Oscillation (ENSO). The characteristics of the upwelling system are set by wind strength related to the South Pacific Anticyclone and the effects of coastal topography on the alongshore wind. The coastal jet off Chile at the eastern flank of the South Pacific Anticyclone (Garreaud and Munoz, 2005;Munoz and Garreaud, 2005) plays also an important role in the feedback mechanism regarding SST conditions and upwelling activity (Renault et al., 2009) .

## 2.2 Global upwelling intensification during the Neogene

Paleo-climatological proxy records off the Western coasts of the continents indicate a cooling for all EBUSs at least since the Pliocene. The strongest cooling is found for the Benguela (-9.3°C) and California (-8.8°C) Upwelling Systems, whereas a weaker cooling (-2.9°C) is found for the Humboldt (Peru and Chile) upwelling region (Dekens et al., 2007).

Benguela upwelling is suggested to have initiated and successively intensified since ~ 14 Ma (Rommerskirchen et al., 2011;Marlow et al., 2000). The Miocene initiation of Benguela upwelling is also assumed to be related to a northward migration of the Angola-Benguela front (Bergh et al., 2018). This is also evident in vegetation changes that indicate an aridification of the adjacent land of Southern Africa (Hoetzel et al., 2015;Dupont et al., 2013).

The Humboldt Current, as well as the Humboldt Upwelling System are suggested to have been active since the early Cenozoic (Keller et al., 1997). Also aridity in the Atacama desert already existed prior to the Miocene/Pliocene (Clarke, 2006). Nevertheless, aridity in the Southern Central Andes is a function of the latitudinal position of the cold upwelling zone along the Chilean coast (Strecker et al., 2007) and hence is influenced by upwelling intensity. According to proxy records the Peru Upwelling System showed a gradual cooling since about 4 Ma (Dekens et al., 2007), whereas the California Upwelling System intensification began in the early middle Miocene (White et al., 1992).

## 3 Uplift histories

All major mountain areas of the world were subject to uplift in the late Neogene. The related processes are likely connected to the activity of two superplumes (Potter and Szatmari, 2009). But studies on the timings of uplift and estimates of uplift rates are rarely in consensus and paleoaltimetry based on stable isotope methods might overestimate rapid uplift, as has been demonstrated in a study for the Andes by Ehlers and Poulsen (2009). Hence the uplift histories of the world's major mountain ranges are still a matter of debate.

### 3.1 Africa

The southern and eastern African plateaus belong to the African superswell (Nyblade and Robinson, 1994). This topographic anomaly is considered to have developed since the early Miocene (Seranne and Anka, 2005)  and it was subject to strong tectonic uplift during the past 30 Myr with a maximum rate in the late Pliocene and early Pleistocene (Moucha and Forte, 2011;Pik, 2011). For Southern Africa, proxy evidence suggests that most of the landscape has been formed during the Neogene, with regional differences ranging from an initiation of uplift in the Namibian region around 30 Ma to more recent uplift intensifications of 10 Ma for the South African swell and the Bié plateau (Roberts and White, 2010). East African uplift can also be attributed to the uplift of rift shoulders that are superimposed on the swell due to volcanism and rifting. Recently evidence for a rather synchronous beginning of uplift of the Eastern and Western parts of East Africa around 25 Ma was found (Roberts et al., 2012), in contrast to a later uplift of the Western part around 5 Ma as previously assumed (Sepulchre et al., 2006). High paleotopography was supposed to have existed in Kenya by at least 13.5 Ma (Wichura et al., 2010), but the remains of a whale found in Kenya well above present-day sea level, dating back to the mid-Miocene constrains the start of the East African Plateau uplift to between 17 Ma and 13.5 Ma (Wichura et al., 2015).

### 3.2 Andes

Late Miocene climatic changes in the Southern Central Andes, with increasingly humid conditions east of the Puna and the development of hyperarid conditions at the Western flanks of the Southern Central Andes is in contrast to the expected aridity due to late Miocene global cooling and hence points to a tectonic origin (Strecker et al., 2007). The Precordillera in the Southern Atacama Desert is suggested to have attained an elevation of at least 3000 m already in the early Oligocene, prior to the uplift of the Altiplano (Bissig and Riquelme, 2010). A recent study of pollen records from western equatorial South America indicated that the Central Andes were already at a reasonable height for blocking air flow and influencing precipitation pattern by the middle Miocene (Grimmer et al., 2018). Most of the Andean uplift is supposed to have occurred during the middle and late Miocene (Potter and Szatmari, 2009). It is suggested that exhumation in the Central Andes moved eastward, beginning in the Eastern Cordillera at ~50-15 Ma, reaching about 75% of present-day elevation in the Bolivian Andes at ~15-11 Ma (Barnes et al., 2012). In the Central Altiplano rapid uplift pulses, that occurred earlier in the South and later in the Northern region, were found to be related to lower lithosphere removal after long-term thickening (Garzione,

2008;Garzione et al., 2014;Leier et al., 2013). Uplift was estimated as 2.5 to 3.5 km between about 10 and 6 Ma (Garzione et al., 2006;Ghosh et al., 2006).

According to (Gregory-Wodzicki, 2000), the Central Andean Cordillera reached about a third of its present altitude 20 Ma ago and about 50% of present elevation by 10.7 Ma implying a magnitude of uplift of around 2300-3400 m since the late Miocene. For the Colombian Andes rapid uplift between 2 and 5 Ma was suggested (Gregory-Wodzicki, 2000). These

timings of uplift of the Central Andes are in agreement with a recorded change in South American seasonality indicating a southward deflection of the South American low-level jet at around 8-9 Ma (Mulch et al., 2010) and with the onset of humid conditions on the Eastern side of the Southern Central Andes in the late Miocene (Strecker et al., 2007). Miocene surface uplift is also documented for the Southern Patagonian Andes (Blisniuk et al., 2005). Uplift of the Bolivian Altiplano was estimated as 2.5 to 3.5 km between about 10 and 6 Ma (Ghosh et al., 2006;Garzione et al., 2006)

**3.3 North American Cordillera**

The North American Cordillera comprises different mountain and plateau areas as the Sierra Nevada, the Rocky Mountains, the High Plains and the Colorado plateau. These have distinct uplift histories with also high levels of uncertainty. Parts of the North American Cordillera were already at a high elevation since the Early Cenozoic (Foster et al., 2010 and references therein). Nevertheless major uplift events took also place from the middle Miocene on. Uplift in the Rocky Mountains

initiated in the middle Miocene, whereas the northern coastal mountains of Canada show a steady rise since 10 Ma (Potter and Szatmari, 2009). The coastal ranges of central and southern California are still rising. They were subject to deformation in the late Miocene, as well as to folding and uplift that initiated around 3.5 Ma (Page et al., 1999;Montgomery, 1993).

**4 Methods**

**4.1 Model Description**

The model simulations were performed with the National Center for Atmospheric Research (NCAR) Community Climate System Model Version 3 (CCSM3) (Collins et al., 2006). The model is run in a fully coupled mode, including the four components atmosphere, land, ocean, and sea ice. The atmospheric component (CAM3) and the land surface model (CLM3) are run at a horizontal resolution of ~1.4° (T85); the atmosphere is discretized with 26 vertical layers. The ocean (POP) and sea-ice models (CSIM) are run with a variable grid resolution of 1.125° in zonal direction and roughly 0.5° in meridional

direction, with a refinement down to 0.3° towards the equator. The ocean model uses 40 levels in the vertical and employs Gent and Mcwilliams (1990) isopycnal mixing for tracers using the skew-flux form (Griffies, 1998). The vertical mixing coefficients are determined using the non-local K-profile parameterization (KPP) scheme by Large et al. (1994). Therefore, surface forcing, including the surface friction velocity which is a function of wind stress, has a direct effect on vertical mixing coefficients. For further details on the ocean model physics the reader is referred to Danabasoglu et al. (2006). The

land model includes a dynamical global vegetation model (DGVM) (Bonan et al., 2003;Levis et al., 2004) and improved

parameterizations for canopy interception and soil evaporation (Oleson et al., 2008;Handiani et al., 2013;Rachmayani et al., 2015). This enables not only the direct simulation of vegetation coverage response to climatic conditions, but also the consideration of biogeophysical feedbacks between climate and vegetation. Clouds are, in CCSM3, implemented as convective or stratiform clouds at three levels (low, middle, high), using different parameterizations dependent on shallow or deep convection. Cloud water is present as ice and liquid, dependent on temperature, explicitly accounting for cloud ice and solid precipitate (Vavrus and Waliser, 2008 and references therein).

## 4.2 Experimental Design

Five model runs were performed. A control simulation (CTRL), as in Jung et al. (2014) which is also described in detail further down, was run using present-day topography. Three sensitivity studies were performed that are identical to CTRL, except that topography was lowered to 50% of the present-day altitude for Eastern and Southern Africa (AF), for the Andes (AND), and for the North American Cordillera (NA), respectively (Figure 1). Additionally a pure preindustrial simulation (PI) was performed for model evaluation purposes.

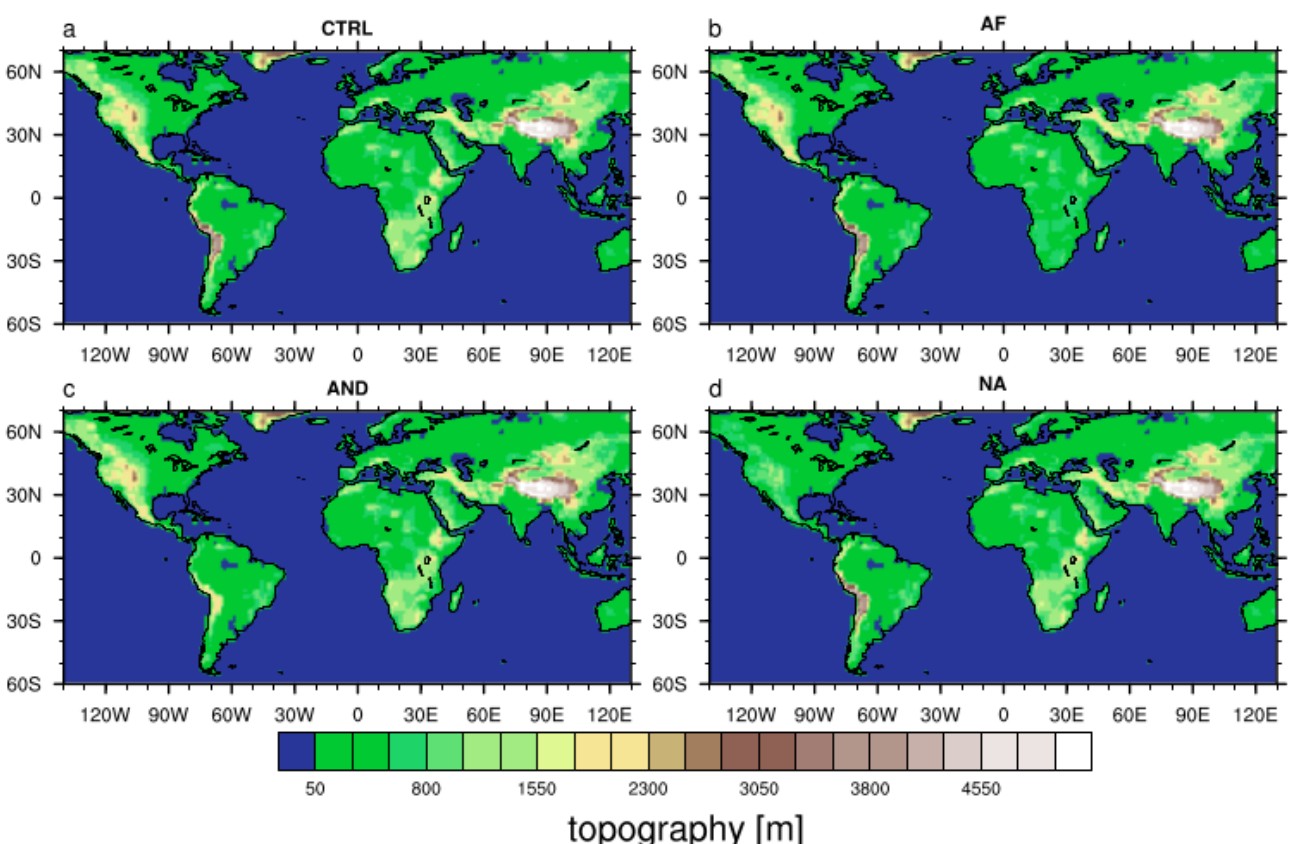

 **Fig. 1: Model topographies of the different experiments: present day topography (CTRL)(a), South and East African topography 50% of CTRL (AF)(b), Andean topography 50% of CTRL (AND)(c), North American Cordilleran topography 50% of CTRL (NA)(d).**

The boundary conditions for the preindustrial simulation (PI) followed the PMIP2 setup (Braconnot et al., 2007). The simulation that serves as a control run in the performed sensitivity experiments (CTRL)has, similarly, present-day boundary conditions with respect to geography, topography and pre-industrial with respect to greenhouse gases, ozone concentrations, sulphate and carbonaceous aerosols (Braconnot et al., 2007), but differs from the pure preindustrial setting in aerosols, soil texture and color and in orbital conditions. Assuming pre-desert conditions, the effect of dust aerosols on radiative transfer was set to zero and soil texture and colour were adapted to represent loam in the Sahara. To avoid extreme insolation forcing, the orbital parameters were set to represent average values over the period of 2-10 Myr with an eccentricity of 0.027 and an obliquity of 23.25°. The perihelion was set to autumn (359.47°). In comparison to PI, our control simulation is characterized by stronger insolation in the northern hemisphere in summer, and weaker insolation during winter, which leads to an increase in seasonality, which is most pronounced in the northern hemisphere. The control run and the preindustrial run were both started from a spun-up preindustrial simulation of 500 years obtained from NCAR and run for another 300 (PI) or 600 (CTRL) years. The sensitivity experiments were branched off from the control run at year 301 and run for another 300 years. It was demonstrated by Jung et al. (2014, Supplementary Information) that the spinup period was sufficient to get the upper ocean in equilibrium. Also the vegetation cover was in equilibrium after the spinup.

All model outputs shown in the following section are averages over a period of 100 years. Differences in the model results section are defined, indicating the anomaly of the control simulation (present-day topography) with respect to the experiment with lowered topography (CTRL minus AF, CTRL minus AND, and CTRL minus NA, respectively). Hence the presented results show the simulated response of the variables to mountain uplift.

## 5 Results

### 5.1 Evaluation of upwelling velocities

For an evaluation of the models' representation of upwelling activity, in the following, the vertical oceanic velocities of all regions are compared to observations, or high resolution modelling results, regarding seasonality and magnitude. Therefore we evaluate the pure preindustrial simulation (PI) which hence differs from the control run (CTRL) especially in the orbital configuration, whereas model geography and topography are identical. Qualitatively, the CTRL run outputs are similar to the PI results illustrated in the following in terms of upwelling velocities and seasonalities (see the appendix for a comparison between PI, CTRL and Carton-Giese SODA 2.2.4 reanalysis).

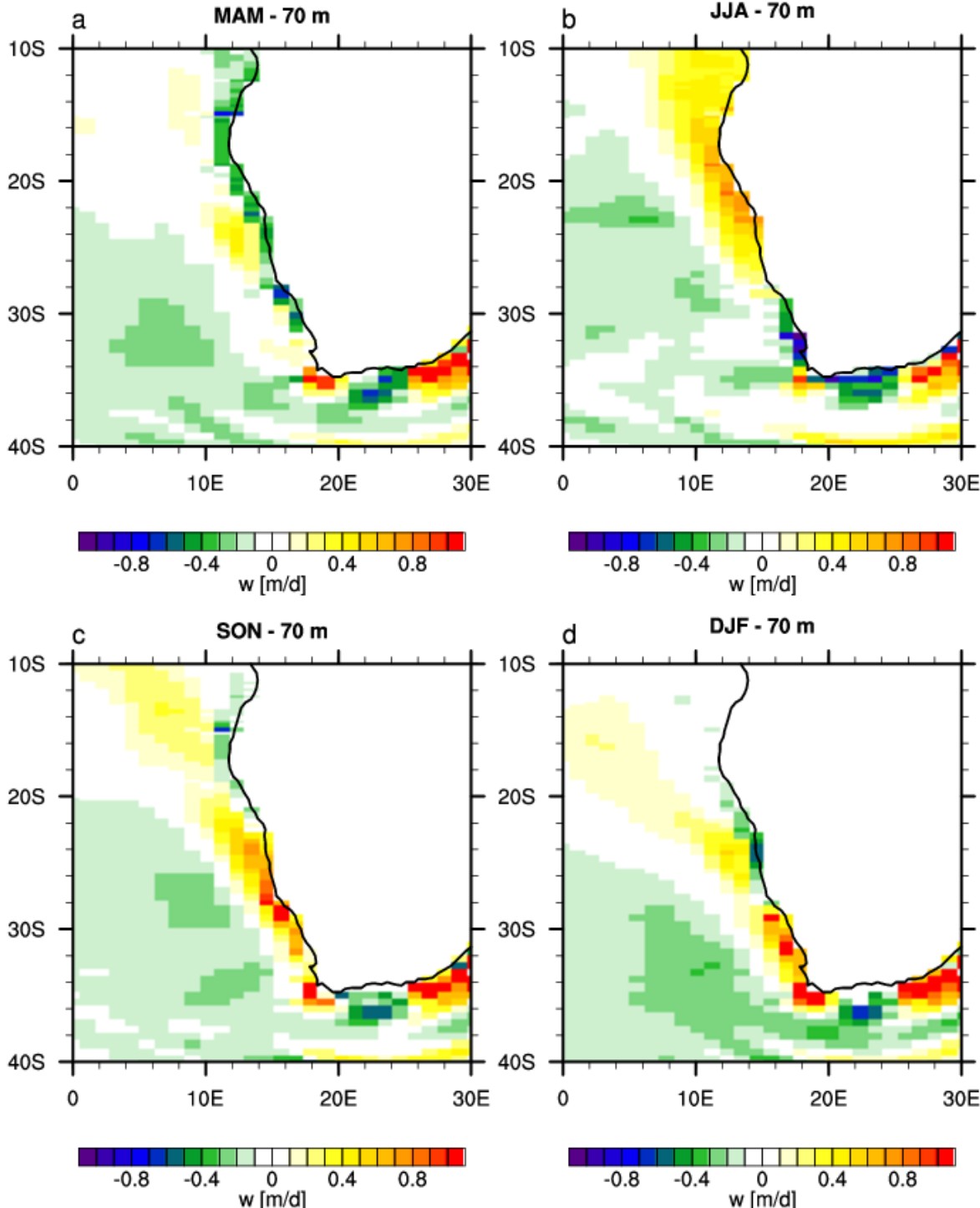

**Fig 2: Seasonal mean vertical velocity [m d$^{-1}$] at a depth of 70 m in a preindustrial simulation (PI) for MAM (a), JJA (b), SON (c), DJF (d). Positive (negative) values denote upward (downward) motion.**

In Jung et al. (2014) it was discussed, based on a comparison between our control run (CTRL) and observations and high-resolutions modeling results, that upwelling location and seasonality are correctly simulated for the Benguela region, but the magnitude of upwelling velocity is underestimated. The same holds true if we consider a run with present-day topography and strictly preindustrial boundary conditions (Figure 2). High resolution model simulations (e.g. Tim et al., 2015;Veitch et al., 2010)   indicate, that the location of the main upwelling zones, as well as their seasonality is captured by CCSM3. We also find a northern and a southern upwelling area with main upwelling in boreal summer and autumn/winter, respectively. In between, in the Central Benguela region we find a region of reduced seasonality. But comparison to these high resolution simulations also shows that the upwelling velocities are generally underestimated by almost one order of magnitude.

For the California upwelling region (Figure 3) the position of the main upwelling zone matches well with the observations (34-44°N; (Chavez and Messie, 2009)). As in Chavez and Messie (2009), upwelling does not occur year-round, but only from spring to summer. In Southern California, where upwelling is weaker, upwelling velocities range from a maximum of 2.5 +-1.3 m d$^{-1}$ in May to a minimum of 0.5+-0.6 m d$^{-1}$ in January (Haskell et al., 2015a). Peak summer vertical velocities of about 10 m d$^{-1}$ were estimated for central Californian upwelling by Pickett and Paduan (2003). Hence, with values mostly below 1 m d$^{-1}$, upwelling velocities in the CCSM3 simulations are underestimated despite showing the correct seasonality.

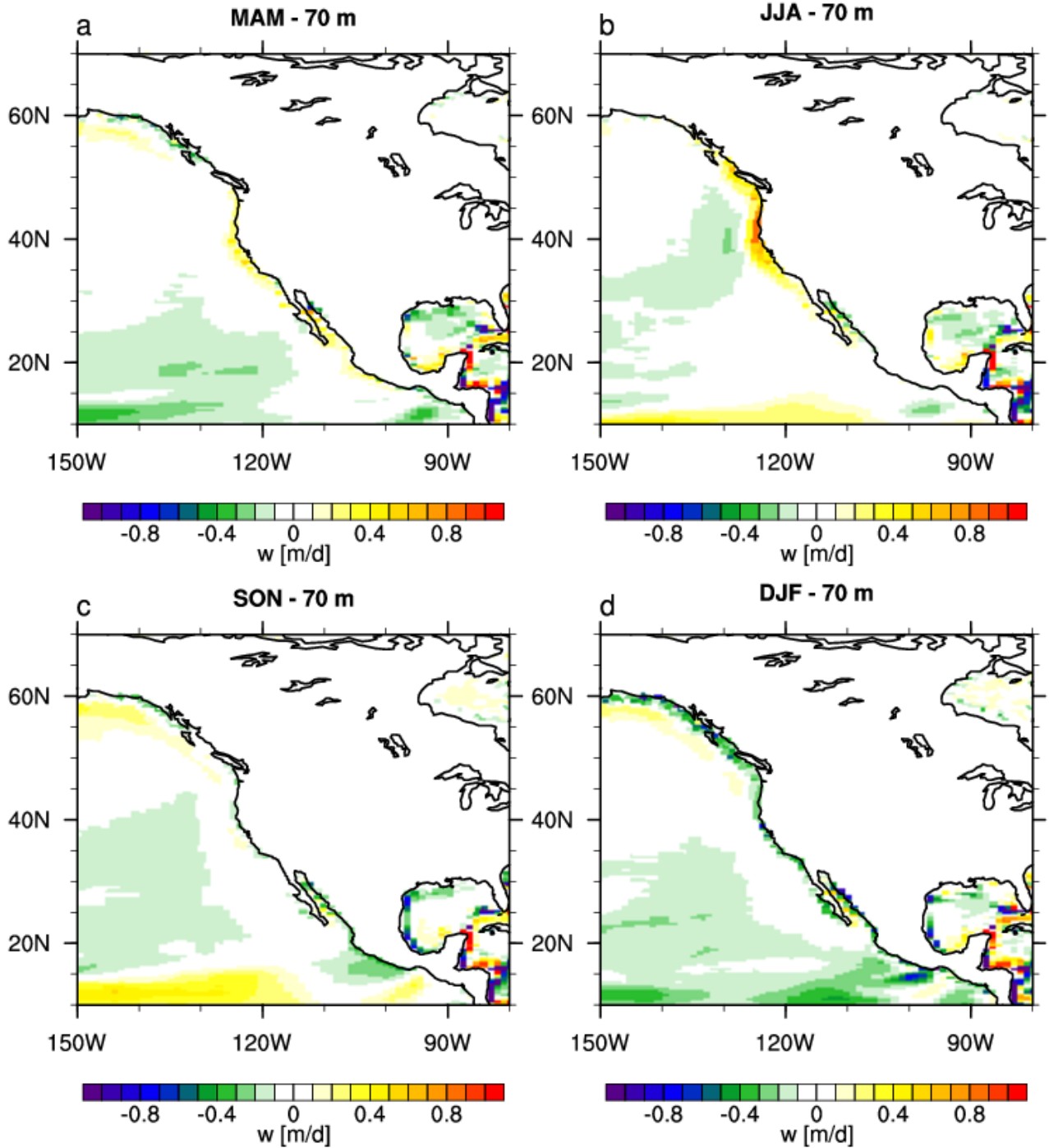

**Fig 3: Seasonal mean vertical velocity [m d⁻¹] at a depth of 70 m in a preindustrial simulation (PI) for MAM (a), JJA (b), SON (c), DJF (d). Positive (negative) values denote upward (downward) motion.**

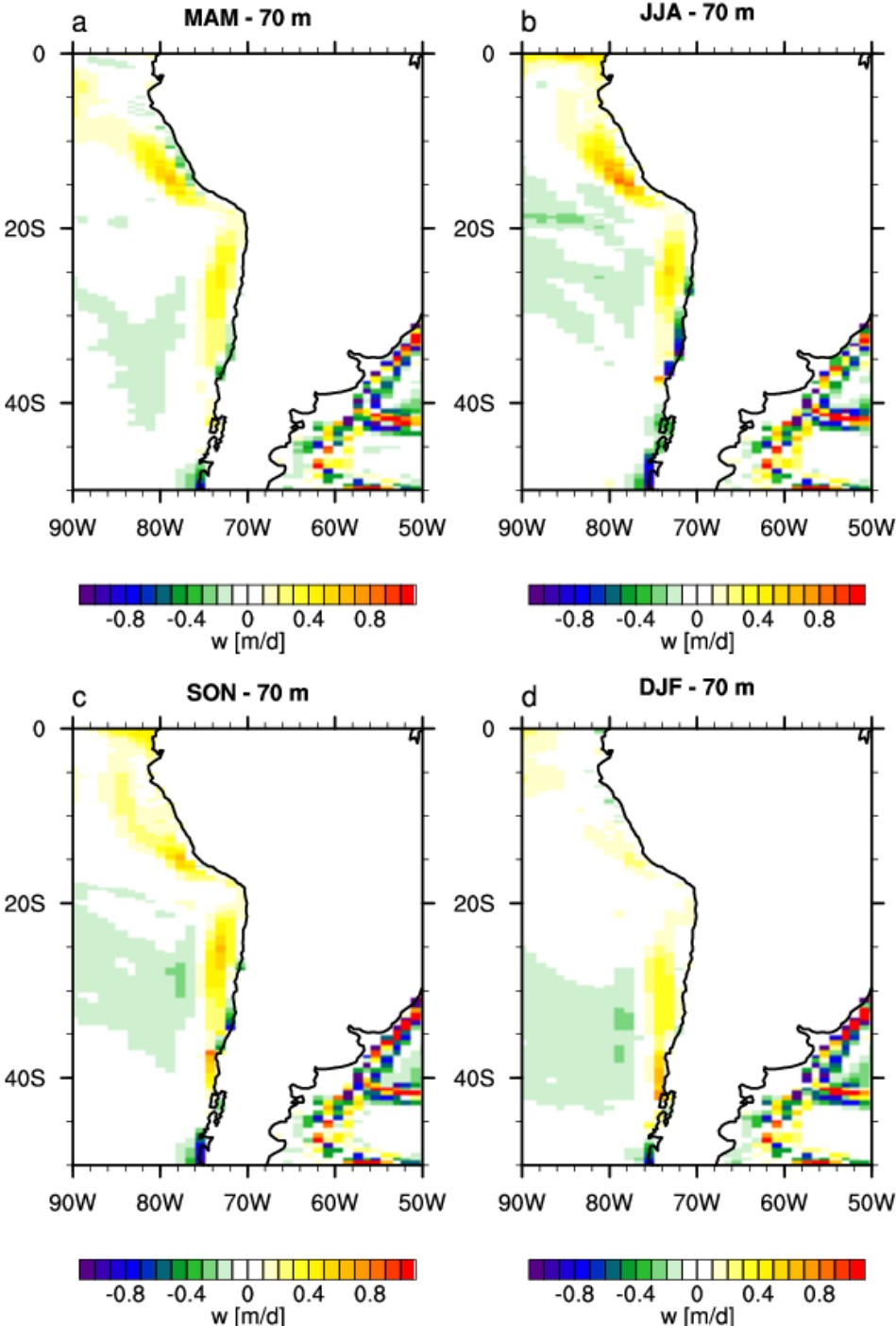


**Fig 4:** Seasonal mean vertical velocity [m d$^{-1}$] at a depth of 70 m in a preindustrial simulation (PI) for MAM (a), JJA (b), SON (c), DJF (d). Positive (negative) values denote upward (downward) motion.

Regarding the Humboldt upwelling region, a comparison of our model results (Figure 4) to higher resolved model results of
Belmadani et al. (2014), using a GCM of variable resolution down to 0.5° over the region of interest, shows reasonably good agreement. Based on observations, upwelling velocities of the eastern tropical South Pacific range from ~ 0.0 to 1.1 m d$^{-1}$, depending on location and season (Haskell et al., 2015b). Therefore our model results compare well with ~0.5 m d$^{-1}$ (up to 0.8 m d$^{-1}$ during JJA in the Peru upwelling region).

Figures A1-A3 in the appendix show seasonal averages of the vertical oceanic velocities of the Carton-Giese SODA 2.2.4
reanalysis data (Carton et al., 2018) for the different upwelling regions. In general the seasonalities are similar to our model results, even though the magnitude of upwelling intensity in our runs is underestimated. In the Benguela region the seasonality is lower in the SODA data, than in the CESM experiments, where upwelling is almost absent in boreal spring.

It was discussed by (Small et al., 2015) that a resolution of about 0.5° and an adjustment to observational data in the coastal zone in the atmospheric model, together with a high resolution eddy-resolving model would be needed to receive a realistic
representation of upwelling for the Benguela region. Ma et al. (2019) also showed that a higher resolution atmospheric component in the coupled model CESM (of ~0.5°) leads to a better representation of oceanic upwelling due to a more realistic representation of the low-level coastal jet. Therefore one has to keep in mind that the effects on upwelling dynamics and feedbacks simulated in this study are likely to be underestimated, as is the general upwelling signal. Even though upwelling is underestimated in all areas, representation of upwelling is sufficient in the chosen model resolution to draw
some conclusions regarding the posed scientific question. Without going into too much detail, since the focus of the study is on Eastern Boundary Upwelling Systems, additional confidence stems from the fact that CCSM3 in the used setup and resolution is also able to reproduce upwelling and its seasonalities in other upwelling areas, e.g. in the Caribbean (Rueda-Roa and Muller-Karger, 2013;Andrade and Barton, 2005;Montoya-Sanchez et al., 2018), and in the South Brazil (Palma and Matano, 2009) coastal areas.

**5.2 Response to mountain uplift**

**5.2.1 Surface wind and Ekman pumping**

For the Benguela region (Figure 5), as well as for the California upwelling region (Figure 6) a clear intensification of low-level winds and Ekman pumping velocities is simulated in our experiments upon uplift.

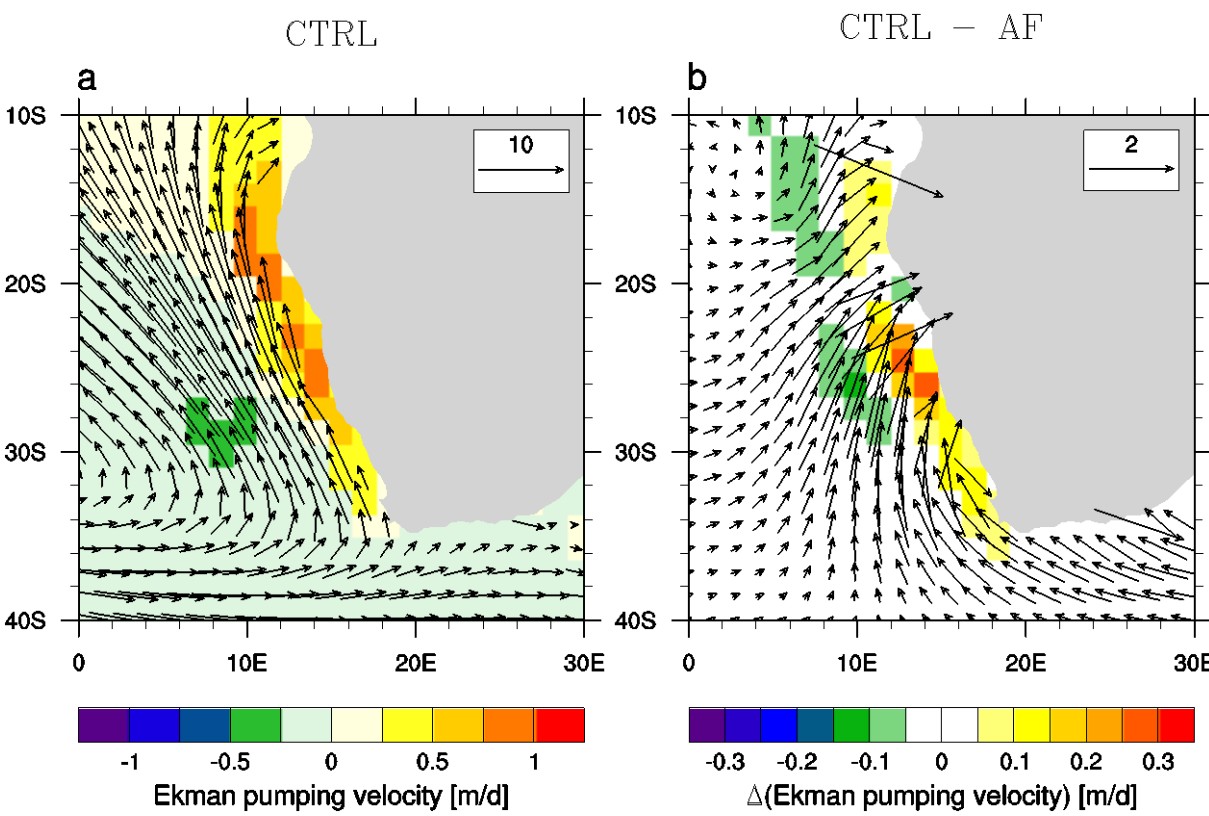

**Fig 5: Annual mean surface wind (arrows) [m s$^{-1}$] and Ekman pumping velocity [m d$^{-1}$] (shading) for CTRL (a) and response to African mountain uplift (CTRL – AF) (b). Positive Ekman pumping velocities imply upwelling.**

For the Benguela region, Ekman pumping is already relatively strong with lower mountains, and further increases by about 60%. For the California upwelling region, the average simulated Ekman pumping intensity is rather weak with low mountains and increases by up to 100% with mountain uplift. Here we illustrate only the summer season for the California upwelling region since most intense upwelling activity is found in that season and upwelling is almost absent in autumn and winter.

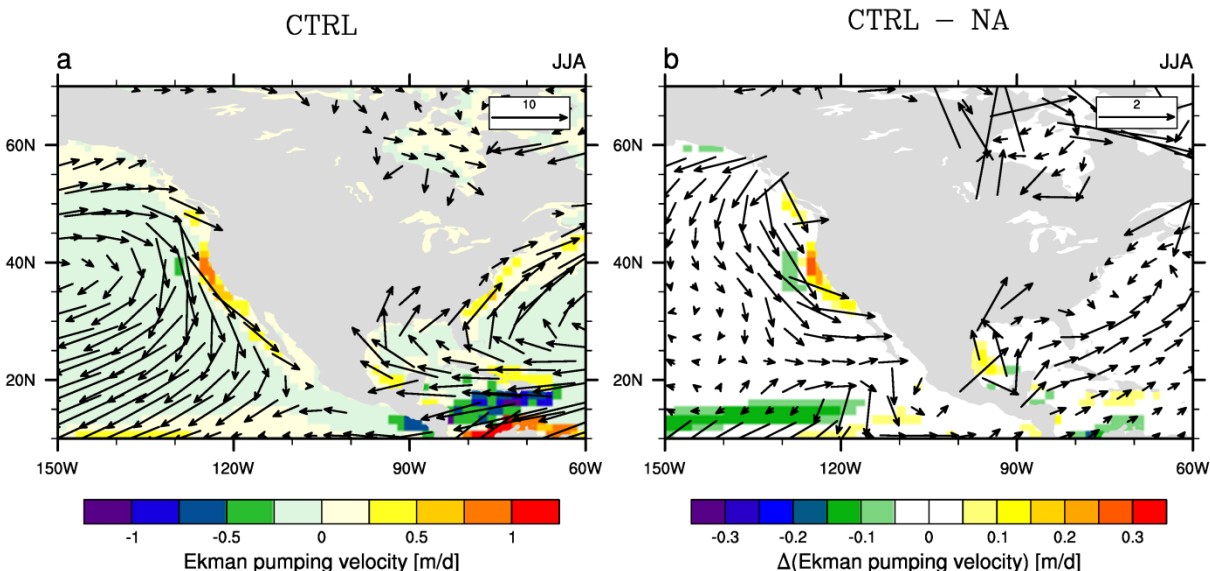

**Fig 6: Seasonal mean surface wind (arrows) [m s⁻¹] and Ekman pumping velocity [m d⁻¹] (shading) for CTRL (a) and response to North American mountain uplift (CTRL – NA) (b), both for the summer season (JJA).**

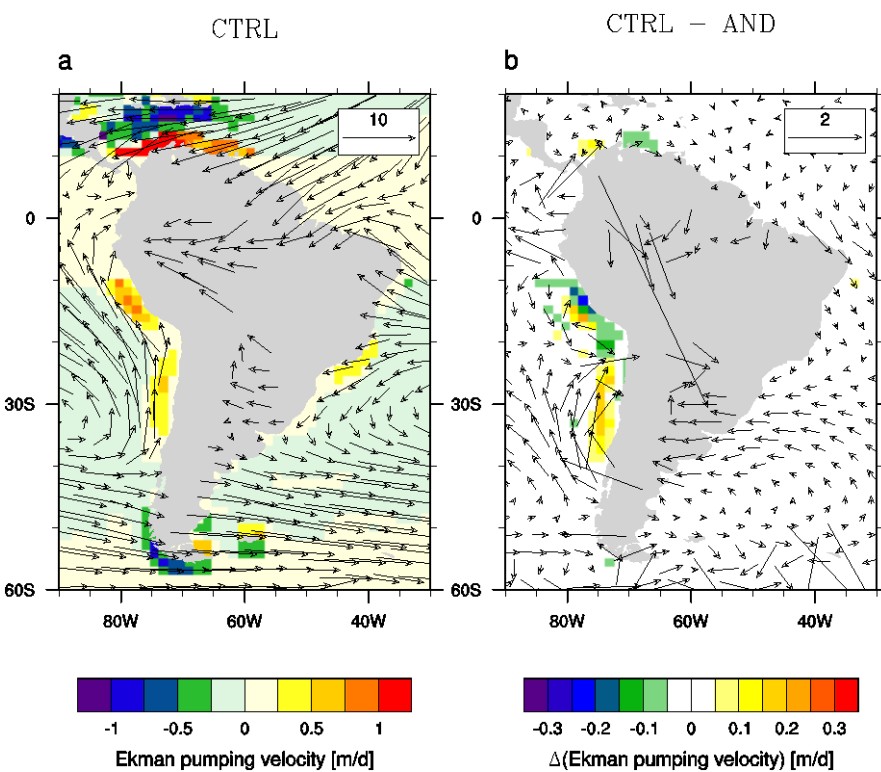

**Fig 7: Annual mean surface wind (arrows) [m s⁻¹] and Ekman pumping velocity [m d⁻¹] (shading) for CTRL (a) and response to South American mountain uplift (CTRL – AND) (b).**

For the Andean uplift experiment (Figure 7), the intensification of wind and Ekman pumping occurs mainly south of ~22.5°S
in the Chile upwelling region, whereas in the annual mean Ekman pumping, as well as wind stress, north of that latitude, in the Peru upwelling region, decreases. For the Southern Humboldt (Chile upwelling) region we find an intensification of the southerly branch of the South Pacific Anticyclone, leading to increased wind stress and Ekman pumping. But not only the alongshore wind component is strengthened; the wind is also directed more towards the coast.

### 5.2.2 Upper-ocean temperatures

Considering changes in SSTs (Figure 8) we find that in the Benguela, as well as in the California  upwelling region there is a cooling of the upper ocean (up to 2.5 °C and up to 1.5°C, respectively). For the Humboldt Current and upwelling region we find a cooling with uplift in the region north of ~22.5°S despite a decrease in upwelling-favourable winds and Ekman pumping intensities. South of that latitude we observe a warming even though Ekman pumping and low-level winds indicate intensified upwelling.

In the following we examine whether the changes in Ekman pumping velocities, that are driven by surface wind modifications are translated into vertical velocity changes in the oceanic column by the ocean model and whether or not this results in a change in the vertical advective heat flux, that is dominating the temperature signal of the different upwelling regions. To this end we illustrate the temperature response together with the vertical motion change of the model layer with the largest response in upwelling velocity for the respective regions (Figure 9).

In the Benguela region zones with increased upwelling velocity clearly coincide with cooling areas (Figure 9a) down to about 150 m (not shown) with temperature responses reaching over 3°C. This was discussed and illustrated also in Jung et al. (2014). In the California region, the match of the spatial distribution of vertical velocity change and temperature change is not as good as in the Benguela Upwelling System. The best overlap is found mostly during boreal summer (Figure 9b) and autumn seasons (not illustrated), the major upwelling seasons of the California Upwelling System. For the Humboldt region

(Figure 9c), this comparison shows that there is no obvious connection between upwelling/vertical heat advection change and temperature change for Andean uplift (neither in the Peru, nor in the Chile upwelling region). Indeed, in the Chile area, despite an increase in upwelling velocities, there is a warming of up to 3°C down to over 200 m depth (not shown).

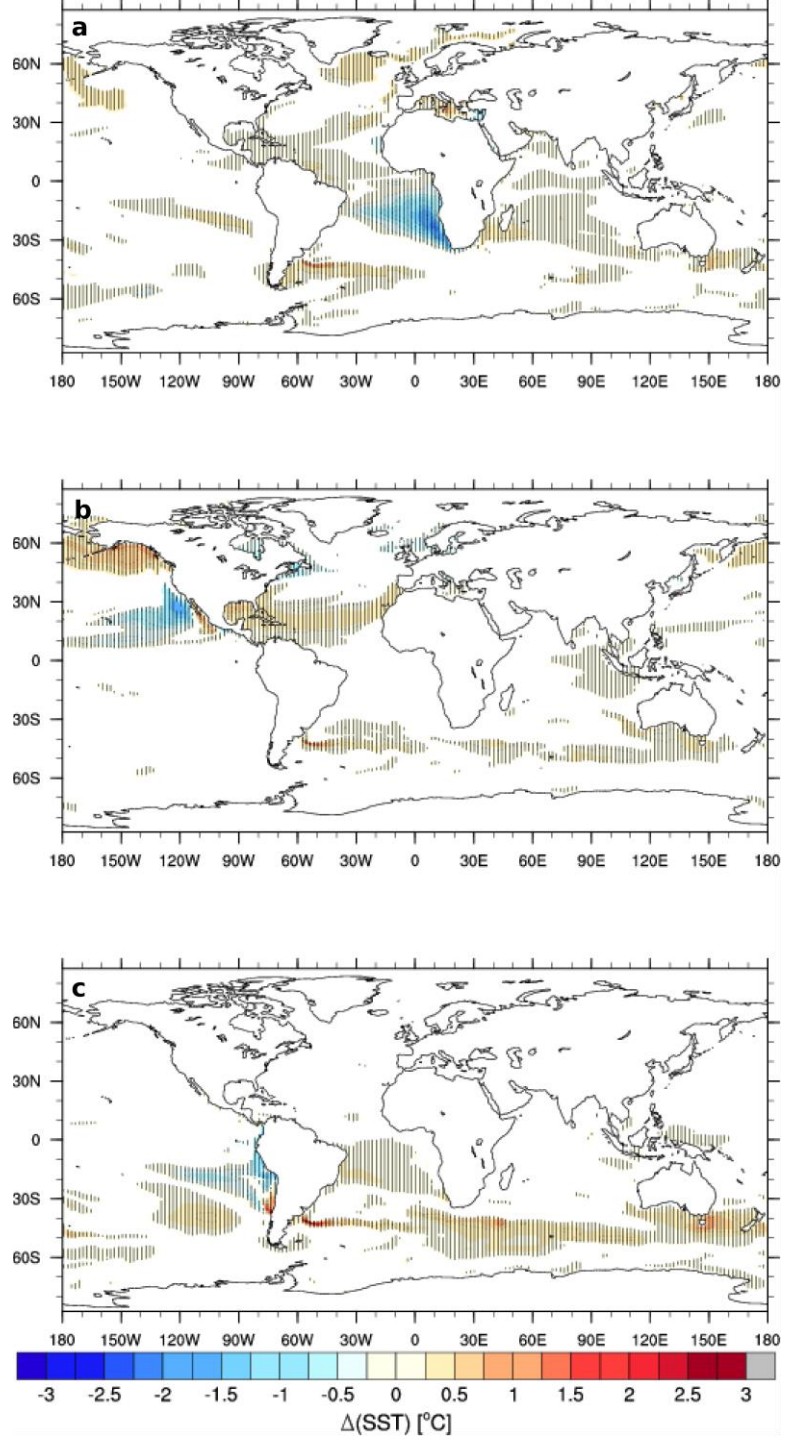


**Fig 8: Annual mean SST response [°C] to African mountain uplift (CTRL – AF)(a), North American uplift (CTRL – NA)(b) and Andean uplift (CTRL – AND)(c), regions of high significance according to t-test (p < 0.01) hatched.**

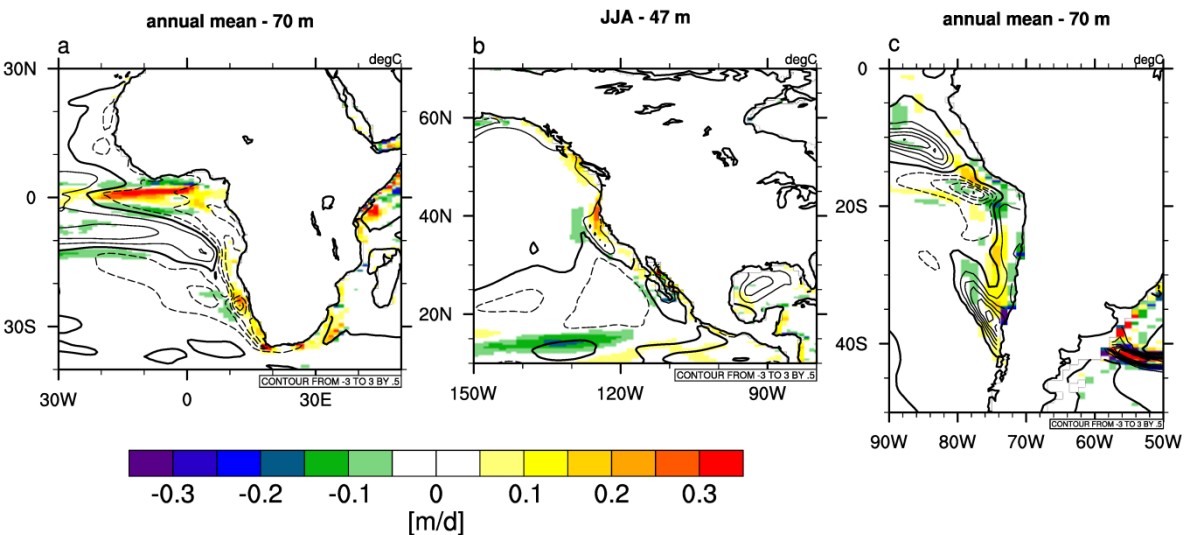

**Fig 9: Vertical velocity responses [m d⁻¹] (shading) and temperature responses [°C] (contours) to African (a), North American (b), and South American (c) mountain uplift. For African (a) and South American uplift (c) the annual mean responses are illustrated for 70 m depth, whereas for North American uplift (b) the response is illustrated for a depth of 47 m (which represents the respective depth of the largest change signal) in the upwelling season in boreal summer (JJA).**

### 5.2.3 Vertical velocity and ocean stratification

Even though there is a clear connection between surface wind strength (and direction) and the intensity of oceanic upwelling, an increase in upwelling velocities does not necessarily lead to an upper-ocean cooling. A second ingredient that is needed is a stratified coastal surface ocean. In case of a very small vertical temperature gradient in the upper ocean, even a large increase in upwelling velocities will not lead to a major cooling.

The following figures illustrate vertical cross-sections at the latitude of maximum upwelling velocities in the coastal zones of the investigated upwelling areas. The cross-sections illustrate vertical velocities and temperatures in the simulation with low mountains and the changes of both variables with respect to mountain uplift. To see a clearer relationship, instead of annual means, we show seasonal averages for the seasons with the largest upwelling intensities and upwelling response intensitites (i.e. SON for the Chile upwelling area and JJA for the other regions).

As discussed in Jung et al. (2014), in the Benguela region, clearly an intensification of upwelling is found that leads to a significant cooling of the upper ocean that is strongest at a depth of ~50-100 m. Here, the upwelling intensification works in concert with the strongest vertical temperature gradient that is found exactly at this depth (Figure 10).

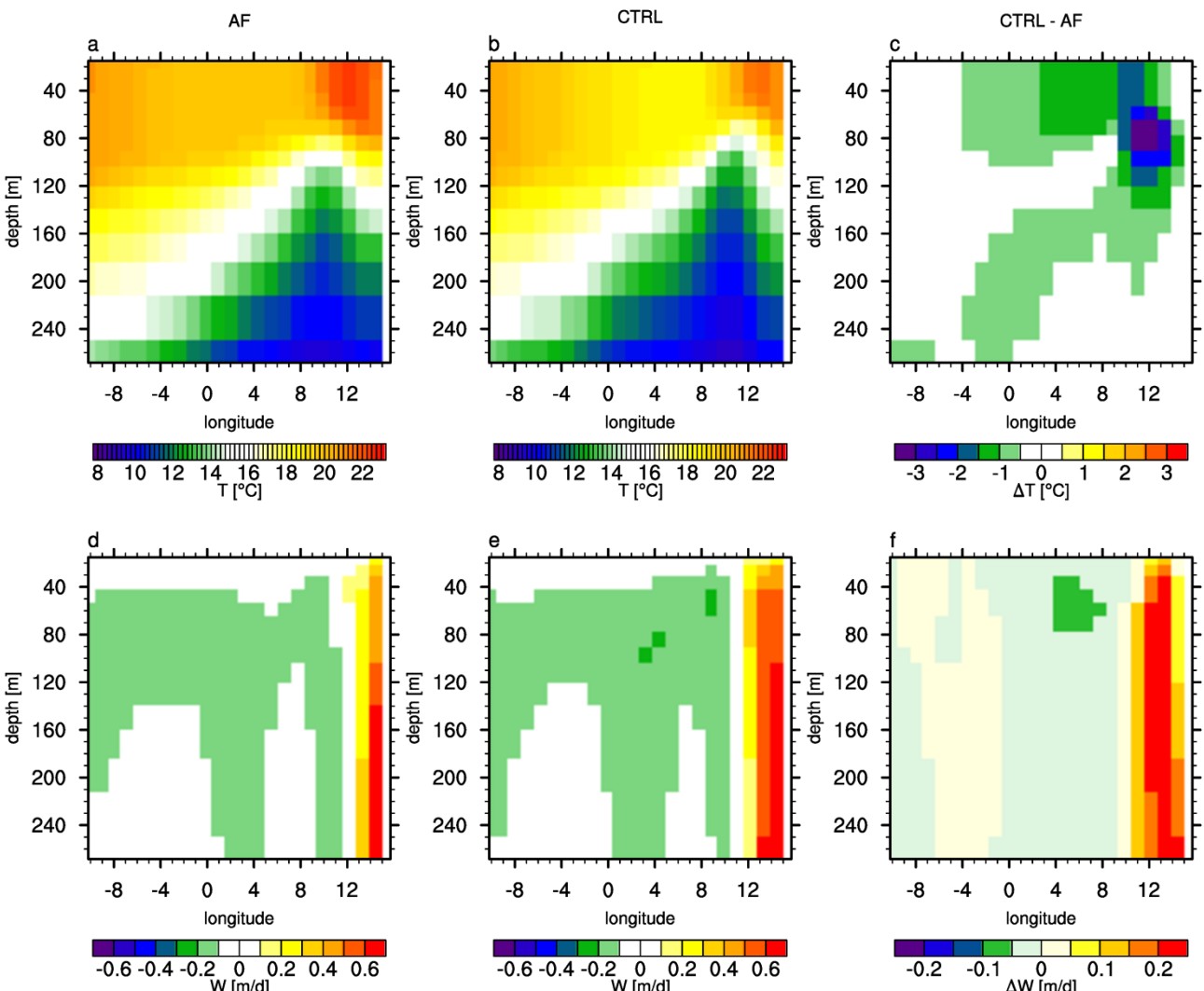

**Fig 10: Vertical cross-sections (at ~25°S): seasonal means of temperature with low topography (a) and high topography (b) temperature response to mountain uplift (c), vertical oceanic velocity with low topography (d) and high topography (e), response in upwelling velocity to mountain uplift (f) for the Benguela upwelling region for boreal summer (JJA).**

The California upwelling region is characterized by a clear intensification of upwelling velocities, which is shown in a vertical cross-section at 39°N (Figure 11, lower panel). Despite this intensification of upwelling velocity, no cooling signal of the upper ocean is found in the region of maximum upwelling response (Figure 11, upper right panel). This is explained by the weak vertical temperature gradient of the surface ocean in that area (Figure 11, upper left panel). The observed surface cooling that is most prominent south of 30°N is hence most likely not mainly influenced by upwelling intensification in this model experiment. In case of a more stratified ocean in combination with a stronger upwelling response (which might

be the case in a higher resolved model and in reality) this might indeed have a larger contribution to upper-ocean cooling, since it was illustrated that simulated upwelling velocities are underestimated.

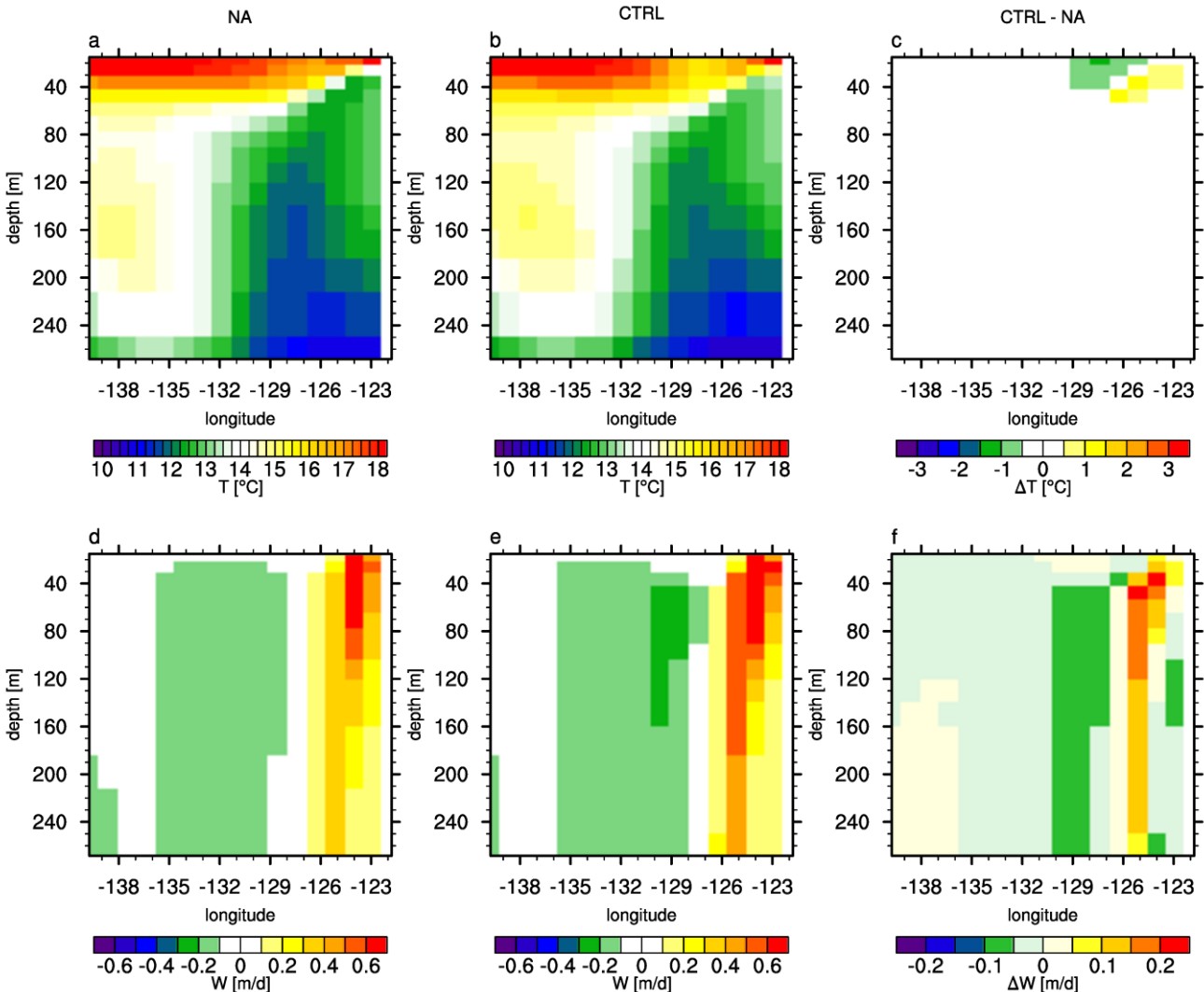

**Fig 11: Vertical cross-sections (at ~39°N): seasonal means of temperature with low topography (a) and high topography (b) temperature response to mountain uplift (c), vertical oceanic velocity with low topography (d) and high topography (e), response in upwelling velocity to mountain uplift (f) for the California upwelling region for boreal summer (JJA).**

In the Peru zone upwelling is evident and we find a small decrease in upwelling intensity in the vicinity of the coast (Figure 12, lower right panel), like it was also evident in Figure 9. Nevertheless we observe a slight surface cooling (Figure 12, upper right panel), even though the stratification of the surface ocean is not minor. Hence other reasons for the temperature change signal with mountain uplift are to be found.

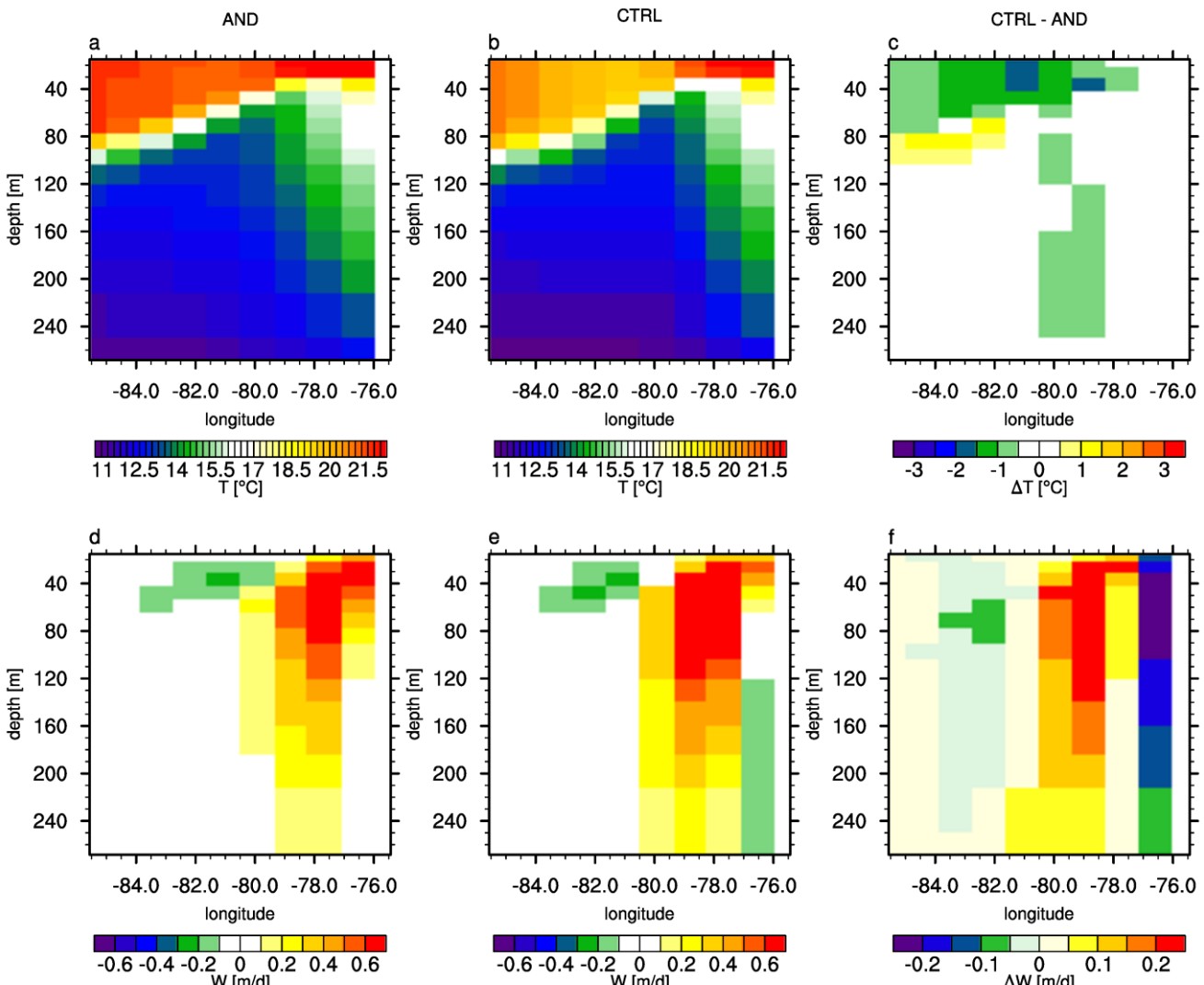

**Fig 12: Vertical cross-sections (at ~15°S): seasonal means of temperature with low topography (a) and high topography (b) temperature response to mountain uplift (c), vertical oceanic velocity with low topography (d) and high topography (e), response in upwelling velocity to mountain uplift (f) for the Northern Humboldt area (Peru upwelling region) for boreal summer (JJA).**

In the Southern Humboldt region, upwelling in the major upwelling zone is rather weak in the model representation for boreal fall (SON) with low mountains, but it increases with mountain uplift (Figure 13). What was not evident from the SST warming in Figure 8 is, that there is indeed a response to the upwelling intensification that manifests in a weak cooling signal at a depth of ~40-80 m. However, upwelling velocities, as well as the ocean stratification are still relatively small with high mountains.

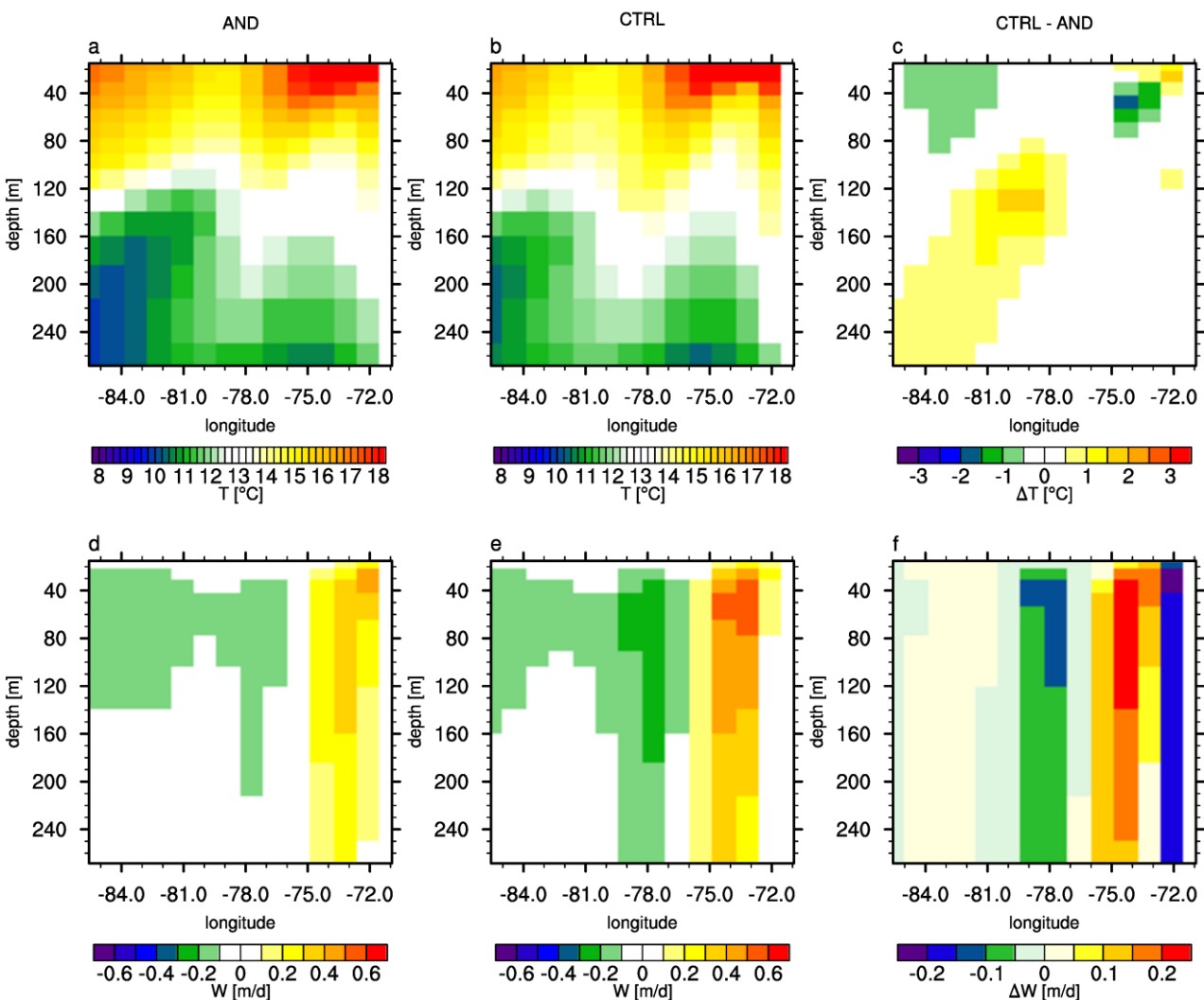

**Fig 13: Vertical cross-sections (at ~29°S): seasonal means of temperature with low topography (a) and high topography (b) temperature response to mountain uplift (c), vertical oceanic velocity with low topography (d) and high topography (e), response in upwelling velocity to mountain uplift (f) for the Southern Humboldt area (Peru upwelling region) for boreal autumn (SON).**

Nevertheless there is a warming signal at the surface and further off the coast down to greater depth (max at ~120 m) that is not related to upwelling activity changes and dominates the response in SST. This temperature change maximum (Figure 8) is found further South (~33°S) than the one illustrated by the cross-section, which is situated at the latitude of maximum upwelling response (~29°S).

## 5.2.4 Surface heat fluxes and low clouds

Surface heat fluxes are likely to alter SSTs in addition to the changes in upwelling strength, but they are also affected by SST changes and hence play a crucial role in atmosphere-ocean feedbacks. In the following we compare surface heat fluxes and illustrate them with respect to their effect on the oceanic heat content. Hence, a reduced solar radiative flux (directed from the atmosphere to the ocean) would lead to a cooling of the ocean, whereas reduced latent and sensible heat fluxes would contribute to a warming of the ocean and hence be displayed as such.

As illustrated in Figure 14 (and similarly in Jung et al. (2014)), in the Benguela upwelling area, the cooling is mainly determined by vertical advective heat fluxes, whereas the surface heat fluxes approximately cancel out. The latent heat flux to the atmosphere is strongly reduced due to reduced SSTs, contributing to a positive change in the surface ocean heat balance. The changes in net surface radiative heat flux and sensible heat flux are negative. Reasons might be an increase in low cloud coverage and the increase in near-coastal wind strength in combination with increased cold-air advection from the South.

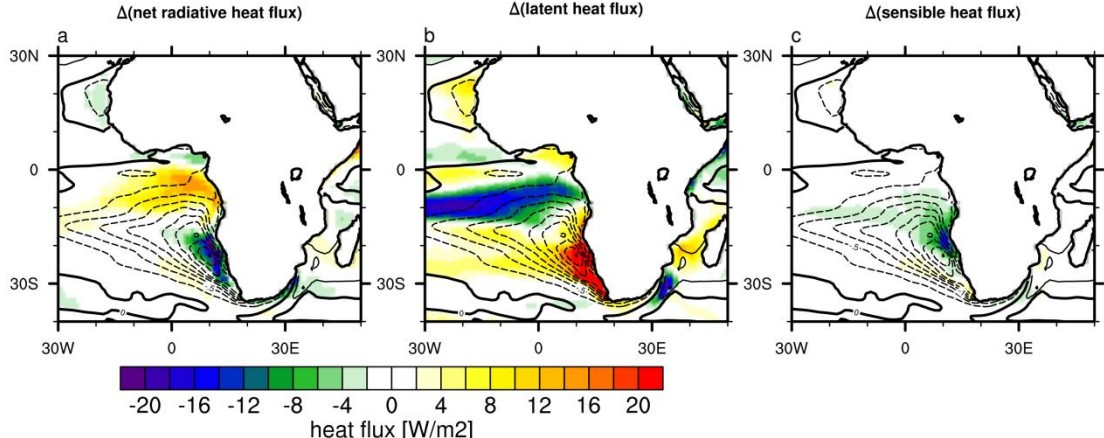

**Fig 14: Annual mean responses of surface heat fluxes to African uplift [W m⁻²]: net radiative heat flux (a), latent heat flux (b), sensible heat flux (c); note: what is illustrated is the effect of the surface heat fluxes on the ocean heat content, contours: SST response (intervals: 0.5°C)**

For the California upwelling region (Figure 15), sensible heat flux changes are also relatively weak. Despite the strengthened surface wind, latent heat flux is slightly weakened due to the surface cooling. Net radiative heat influx is strongly decreased due to increased low cloud coverage, as is illustrated below. The net radiative heat flux change is by far the most important contribution of the surface fluxes and hence acts in concert with the rather weak upwelling intensification.

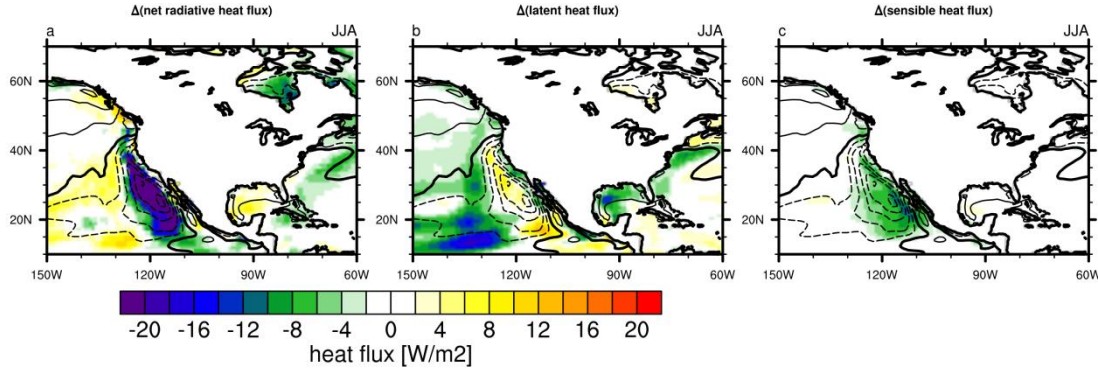

**Fig 15: Seasonal mean (JJA) responses of surface heat fluxes to North American uplift [W m⁻²] in boreal summer: net radiative heat flux (a), latent heat flux (b), sensible heat flux (c); note: what is illustrated is the effect of the surface heat fluxes on the ocean heat content, contours: SST response (intervals: 0.5°C)**

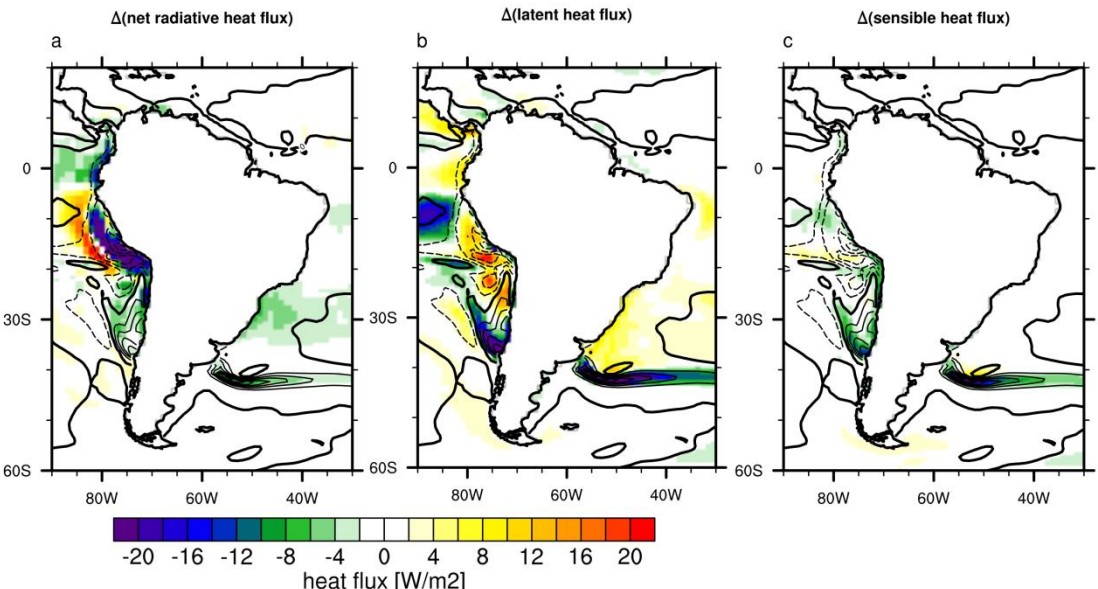


**Fig 16: Annual mean responses of surface heat fluxes to Andean uplift [W m⁻²]: net radiative heat flux (a), latent heat flux (b), sensible heat flux (c); note: what is illustrated is the effect of the surface heat fluxes on the ocean heat content, contours: SST response (intervals: 0.5°C)**

In the Northern Humboldt region the cloud radiative feedback acts to reduce surface energy balance by altering the net

radiative heat flux (Figure 16) and hence opposes the reduction of upwelling which would otherwise lead to a warming of

the sea surface. The latent heat flux is reduced, most likely as a consequence of the decrease in surface winds and of surface

cooling and hence counteracts the overall cooling. Sensible heat flux changes are small for the Northern Humboldt region

(and of different sign) and therefore have a negligible effect on SSTs. From this analysis it appears that the strongest effect

of mountain uplift works via the net radiative heat flux, leading to a cooling, despite a reduction in upwelling and the

contrary effect of latent heat flux. This reduction of the net radiative heat flux is mainly caused by an increase in cloud coverage, which will be discussed in the following.

In the Southern Humboldt region both, net radiative and sensible heat flux are not of major importance for SST change, whereas the latent heat flux is strongly increased (indicated by a negative contribution to the surface heat balance of the ocean). Most likely this effect is caused by the increase in SST and the intensification of the wind strength. All components

of the surface heat balance indicate a cooling in the Southern Humboldt region and are hence acting in concert with upwelling intensification. Therefore other explanations are needed for the strong warming of the surface ocean that can be observed in the Southern Humboldt upwelling area.

In three of the upwelling areas, namely the Peru, Benguela and California upwelling zones we could identify a strong decrease in net radiative heat flux. This net radiative heat flux response is to a major part caused by an increase in low cloud

coverage that is illustrated in Figure 17a-c for the different experiments, impacting shortwave incoming radiation. But there is also some contribution from a decrease in higher level cloud cover. This is also shown by an analysis of the shortwave and longwave contributions to the radiative feedback (compare figures A7-A9 in the supplement). In the Benguela upwelling region the reduction in shortwave insolation is caused mainly by an increase in low cloud coverage. Additionally, in the northern part of the Benguela region, the net radiative cooling is enhanced by a decrease in mid-level and high clouds, which

leads to a reduced longwave cloud forcing. In the North American area, the largest part of the cooling signal can be traced back to an increase in low clouds and an additional cooling in the southernmost area reaching far out in the ocean can also be attributed to a decrease in mid-level and high clouds. In the Southern Humboldt region, there is no effect of cloud radiative forcing, but in the Northern Humboldt region shortwave cloud radiative forcing due to increased low cloud cover leads to surface ocean cooling. Longwave cloud forcing is, in the Andean uplift experiment only relevant over the continent.


.

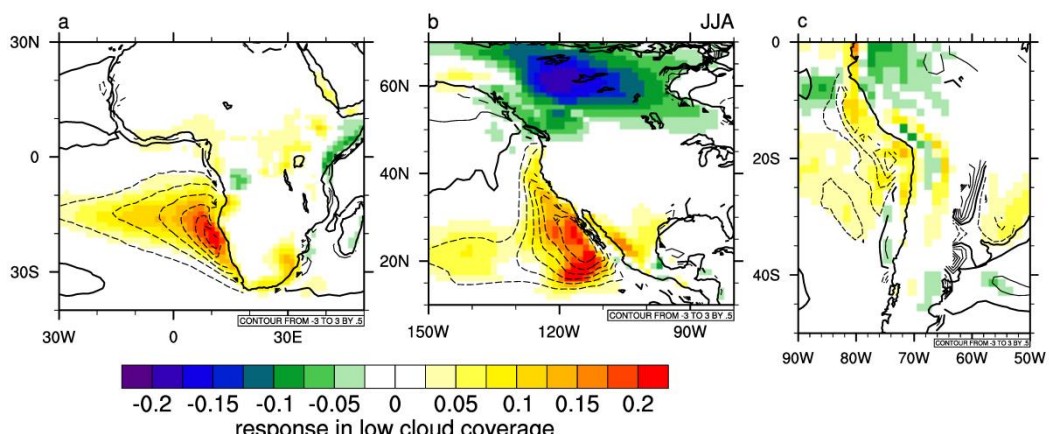

**Fig 17: Responses in low cloud coverage (shading) and temperature (contours, intervals: 0.5°C) to African (CTRL – AF)(a), North American (CTRL – NA)(b), and South American (CTRL – AND)(c) mountain uplift. For African (a) and South American uplift (c) the annual mean responses are illustrated, whereas for North American uplift (b) the response is illustrated for the upwelling season in boreal summer (JJA).**

## 5.2.5 Horizontal heat advection in the surface ocean

Another important factor related to oceanic temperature evolution is the influence of horizontal heat transports. In the Benguela region (Figures 18a and 18d), there is mainly a strengthening of north-westward current velocities and hence advection of colder waters from the Southeast occurs. The flow pattern also indicates that the cooling signal is mainly connected to the region of maximum upwelling intensification and is eventually transported with the prevailing flow pattern. The cooling is additionally intensified by the increase in the transport of cooler waters from the South.

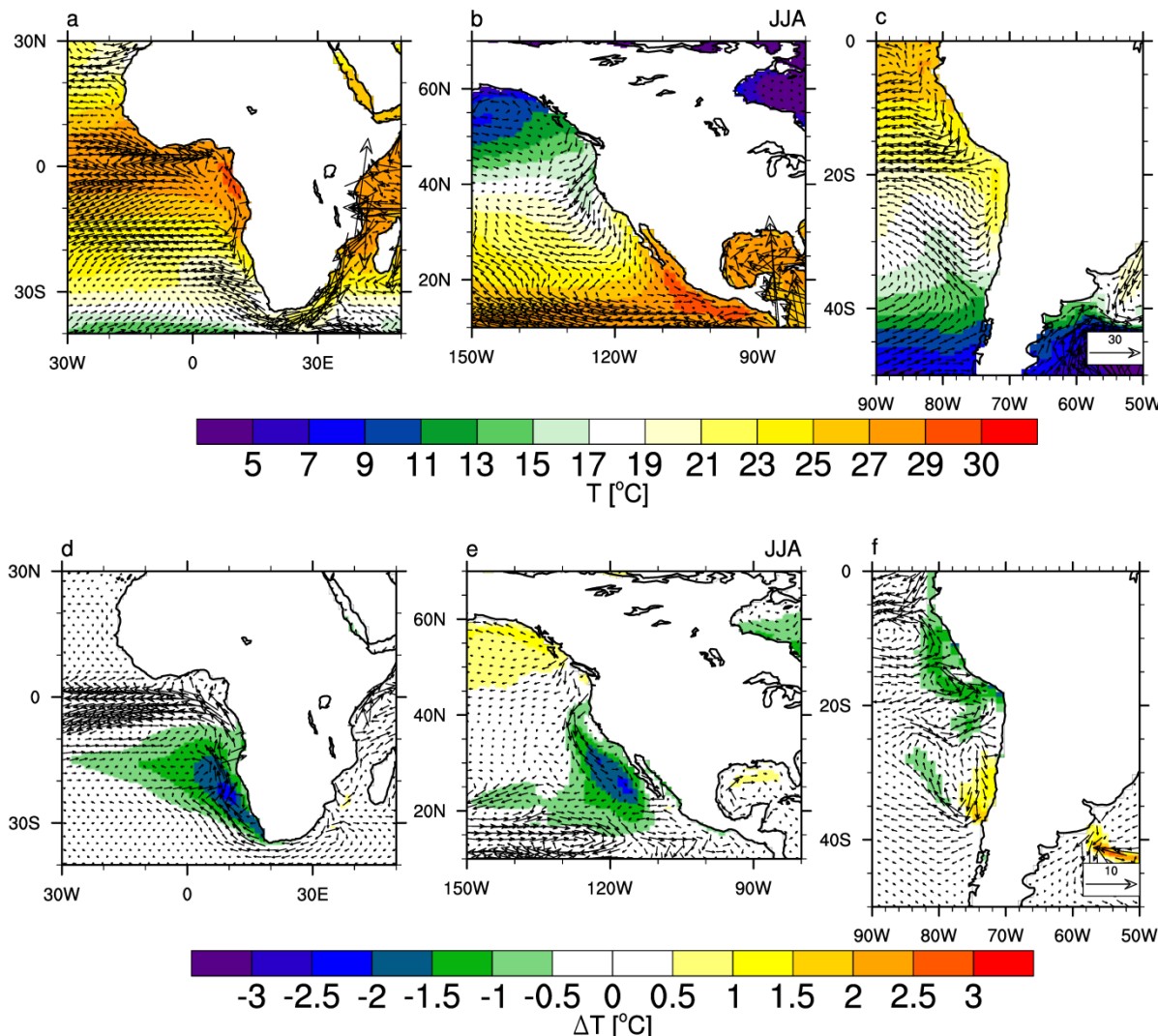

 **Fig. 18: Surface (first model layer) horizontal velocity [m s⁻¹](vectors) and SST [°C](shading) for CTRL (a,b,c); horizontal velocity responses [m s⁻¹] (vectors) and temperature responses [°C] (shading) to African (CTRL – AF)(d), North American (CTRL – NA)(e), and South American (CTRL – AND)(f) mountain uplift. For Africa (a,d) and South America (c,f) the annual means are illustrated, whereas for North America (b,d) the seasonal mean in the upwelling season in boreal summer (JJA) is shown.**

In the California region (Figures 18b and 18e), the strong cooling signal seems to be most likely related to the strengthening

of the southward transport of cool water masses. Here, the strongest cooling signal is not found in the area, where we see the strongest upwelling intensification (Figure 9), but further to the South, where the southward flow intensification ceases and turns into a slight southward flow decrease. This indicates that in contrast to the Benguela upwelling region, upwelling does not play an as important role for upper-ocean cooling, whereas horizontal heat transport is more important.

For the Peru upwelling zone (Figures 18c and 18f) there seems to be no direct connection between the transport of water

masses and SST, since the horizontal temperature gradient in that area is very small.

In the Chile upwelling zone (Figures 18c and 18f), where surface energy fluxes, as well as upwelling intensities did not show any relation to the strong upper-ocean warming, we can clearly relate the temperature changes to an intensification of the Peru-Chile undercurrent and of the Peru-Chile countercurrent. This southward flow anomaly is most likely related to changes in wind stress curl and associated Sverdrup dynamics (Karstensen and Ulloa, 2009). As illustrated in the cross-

section in Figure 13, also an upwelling-induced small cooling signal is found in the sub-surface, but this is overlain by this very strong transport of warm water masses from the North to the South.

## 6 Discussion

Dekens et al. (2007) suggested that the long-term cooling trends, which are obvious in all major EBUSs since at least the Pliocene, are attributable to factors like changes in depth and/or temperature of the thermocline. Even though these factors

might have played a role (Steph et al., 2010) our study suggests that mountain uplift processes need also to be considered as a cause contributing to the cooling in some of the EBUSs.

Not many studies were performed so far, analysing the effect of mountain uplift on atmospheric and oceanic dynamics with a coupled global general circulation model. Jung et al. (2014, 2016) illustrated how African topography shaped atmospheric circulation patterns that have a strong influence on the strength of the Benguela Upwelling System, but also on precipitation

and vegetation cover. Sepulchre et al. (2009) investigated the influence of Andean uplift on the Humboldt Current. They found rather similar results regarding the atmospheric response with their uncoupled atmosphere general circulation model (AGCM). The oceanic response comparing our coupled model to their offline used regional ocean circulation model ROMS shows large differences, most likely caused by missing feedback mechanisms and the higher resolution of the regional ocean model. A model study, applying the relatively highly resolved atmosphere-only GCM CAM5 for present-day and Pliocene

topography (globally) to investigate the effects of uplift on the worlds' major upwelling regimes showed similar results in terms of the cloud-radiative response to mountain uplift (Arnold and Tziperman, 2016). Potter et al. (2017) compared a low resolution atmosphere-only and a coupled model for the response to Andean uplift. Their simulation also showed an

increased low cloud formation with high mountains and a much reduced cloud coverage in the low mountains scenario. The low cloud reduction is stronger in the non-coupled simulation, but extends further West over the ocean in the coupled

simulation, due to atmosphere-ocean feedbacks and dynamic ocean processes, as well as model deficiencies.

Model experiments for South America and Africa (Richter and Mechoso, 2006, 2004;Feng and Poulsen, 2014) showed that elevated terrain acts on lower tropospheric stability, resulting in an increase of low cloud coverage off the coast. This is caused by the blocking and deflection of the Westerlies (for South America additionally a blocking of the North-Easterly flow) which leads to subsidence warming above the planetary boundary layer and cold-air advection within the planetary

boundary layer. This increase in cloud coverage already occurs in the studies without coupled ocean (Richter and Mechoso, 2006, 2004), but the effect is most likely strengthened by atmosphere-ocean feedback mechanisms, since the increased cloud coverage leads to a sea surface cooling due to reduction of incoming solar radiation which then leads to an enhanced cooling in the lower layers of the atmosphere which further enhances the effect on tropospheric stability and low-level cloud coverage, forming a positive feedback loop. In the study of Feng and Poulsen (2014)), using the coupled global circulation

model CCSM4, the radiative cooling caused by low clouds even leads to a remarkable reduction of the Walker circulation and a reduced ENSO strength and frequency due to the cooling of the eastern equatorial Pacific.  In terms of clouds, surface wind, SST and also horizontal heat advection a simulation with flat Andes compared to present-day orography showed results similar to our simulation. Another experiment with moderate half Andean elevation resulted in only minor responses. The main difference between their study and the present investigation is, that they did not show an opposite effect for the

Peru and the Chile upwelling areas, which might be related to a lower model resolution. The similarity of the different studies underlines the robustness of our findings.

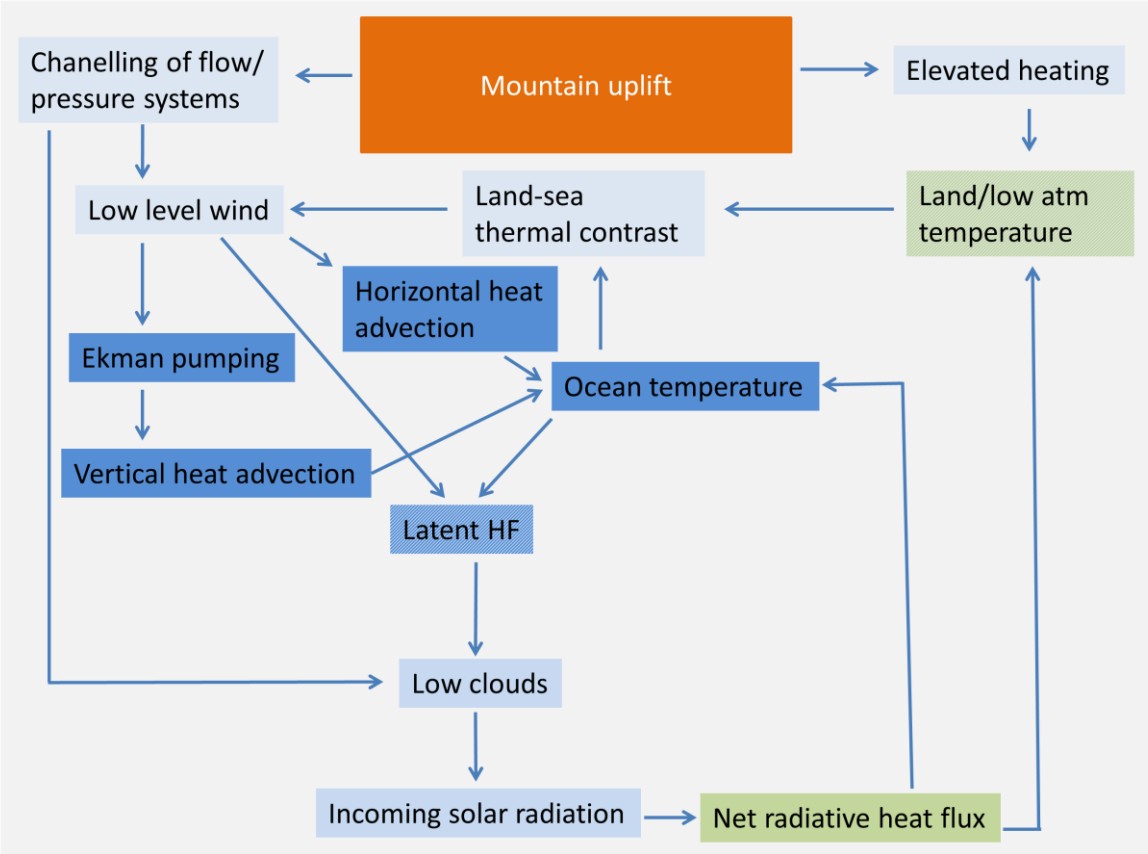

**Fig. 19: Summary of relevant processes interacting in case of continental-scale mountain uplift considered in this study. The complexity of the processes lead to negative, as well as positive feedback mechanisms that also depend on the regions' characteristics. Colors: light blue – atmospheric processes, blue – oceanic processes, green – land surface related processes, orange – forcing (i.e. mountain uplift).**

Figure 19 illustrates the most important processes that were analysed in this study and that play a role in the responses of EBUSs to continental-scale mountain uplift. Mountain uplift on one hand leads to elevated heating which then results in a strengthened land-ocean temperature gradient in the lower atmosphere, which through thermal wind relation leads to intensified low-level jets that are the main drivers of upwelling in EBUSs. Additionally, higher mountains directly affect the low-level winds near the coast by a channelling of the flow. However, the low-level winds do not only affect upwelling activity, but also act on horizontal heat transport which might lead to either a warming, or a cooling of upper-ocean temperatures. Low-level winds, as well as ocean temperatures, act on the latent heat flux which is an ingredient for the formation of low clouds. More importantly low cloud coverage is influenced by mountains due to blocking of westerly flow and an increase in lower tropospheric stability, as described in detail by Richter and Mechoso (2006, 2004). The increase in low cloud coverage reduces downward shortwave radiation, and hence the net radiative heat flux, leading to an additional

cooling of the sea surface. The resulting SSTs then also form part of the atmosphere-ocean feedback in EBUSs, as described in Nicholson (2010) for the Benguela upwelling region.

Nevertheless one has to be aware of model deficiencies that might have a strong impact on the effect of boundary condition changes on climate dynamics, as well as on the magnitude of different effects and feedbacks.

According to Kiehl et al. (2006) CCSM3 shows an enhanced low-cloud feedback due to deficiencies in the simulation of boundary layer mixing processes with enhanced low cloud cover in their control simulation. Hence the ocean-atmosphere feedback might be overestimated. On the other hand, the intensity of coastal jet, upwelling intensity and hence also the changes in those variables might be underestimated due to a too coarse model grid (see Section 5.1). Proxy records of the major EBUSs indicate a SST cooling of around 3-10°C (Marlow et al., 2000;Dekens et al., 2007;Rommerskirchen et al., 2011;Miller and Tziperman, 2017), which is larger than the cooling simulated in our model experiments (1-3°C). Hence mountain uplift is most likely only one piece of the puzzle to Miocene/Pliocene cooling of Eastern Boundary Upwelling Systems, but its contribution might be larger in reality due to the aforementioned model deficiencies.

**7 Summary and Conclusions**

In almost all studied regions with coastal mountains and prevailing coastal parallel winds (Benguela, California and Southern Humboldt), we find intensified low-level winds, and an intensification of Ekman pumping velocities with mountain uplift, most likely related to the thermal wind relationship, following elevated heating, as explained in Jung et al. (2014). To which extent this leads to SST cooling depends on a variety of influencing factors and feedback mechanisms that differ for the different upwelling regions.

- Benguela region: the coastal low-level jet and Ekman pumping intensification are the dominant factors in controlling the SST response, but horizontal heat advection also contributes to upper-ocean cooling.
- California region: the coastal low-level jet and Ekman pumping intensify during the summer season, but horizontal heat advection, as well as the net radiative effect due to an increased low cloud coverage are likely more important for SST cooling.
- Humboldt region: the weak ocean stratification hinders SST response to changing upwelling velocities. In the Chile upwelling area horizontal heat advection is dominant. Upper-ocean warming is observed, despite an intensified coastal low-level jet and strengthened upwelling activity. In the Peru upwelling zone the net radiative cooling effect is dominant, which is also caused by a low-level cloud coverage increase.

It could be demonstrated, that mountain uplift is important for upper-ocean temperature evolution in the area of Eastern Boundary Currents. The relative importance of the different feedbacks and processes depends on the upwelling region (e.g. mountain elevation, ocean stratification), but might also be influenced by model resolution, since the response of upwelling to low-level wind intensification in a global GCM is limited and most likely underestimated.

## Author contributions

Gerlinde Jung and Matthias Prange designed the experiments. Gerlinde Jung set up the model, performed the simulations
and the output analysis and prepared the manuscript with contributions from Matthias Prange.

## Acknowledgements

This work was funded by the Deutsche Forschungsgemeinschaft (DFG) Research Center/Excellence Cluster "The Ocean in the Earth System".

The authors like to thank Gary Strand of NCAR for making available the spun-up preindustrial control run restart files and
Matthew Hecht from the Los Alamos National Lab for his assistance with the production of mapping files for this model version. Thanks to Bruce Briegleb (NCAR) for making pre-industrial aerosol data available.

The CCSM3 experiments were run on the SGI Altix Supercomputer of the "Norddeutscher Verbund für Hoch- und Höchstleistungsrechnen" (HLRN). Model setup and output analysis and plotting was performed using cran R (R-Development-Core-Team, 2011) and NCL software.

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
