# Peer review of "The effect of mountain uplift on eastern boundary currents and upwelling systems"

_Climate of the Past, 2019_

## Referee Comment (RC1) · Anonymous Referee #1 · 12 May 2019

Modern coastal upwelling systems initiated and intensified since the Neogene. However, the reasons for their strengthening throughout the Miocene and Pliocene remain unclear. In the paper, the authors carry out sensitive experiments to investigate the impacts of mountain uplift on the three upwelling systems. The authors carefully diagnose the model outputs, in particular clearly illustrate the feedbacks behind the upwelling responses. The paper is well written. I would recommend its publication after considering the suggestions below.

General comments:

1. The author should introduce the vertical mixing schemes in the model. In addition to the background vertical mixing, does the model include other vertical mixing parameterizations, for example the tidal mixing, the eddy mixing. Some of these mixings are

also influenced by changes in winds. In other word, when the topography is modified, the changes in winds will also influence these vertical mixings. If these vertical mixings remain unchanged, there are some uncertainties included in the current simulations.

2. The uplifts of the Andes and North American Cordillera induce significant cooling around the adjacent upwelling regions. The authors should potentially compare some model outputs with existed proxy data?

3. For the cross-section analysis, I recommend the authors could also do that with an averaged latitude zone over the upwelling regions, especially for the vertical velocity response, rather than using a specific latitude.

4. I am interested in the thermocline depths changes around the three upwelling regions and their potential impacts on the cooling strength.

Specific comments:

1. Page 3 line 15 : "Neogene" not "Neogen"

2. Figure 1 and 9-12, Please denote each panel with alphabet letters.

3. Figure 8b, please explain why choose the depth of 47m here rather than 70m?

---

## Referee Comment (RC2) · Anonymous Referee #2 · 26 Jun 2019

This manuscript explores the impact of different mountain uplifts on eastern boundary upwelling systems, through a set of sensitivity experiments to topography run with CCSM3 model. It echoes a previous publication by the authors, but this particular ms appears as a generalization assessment of the previous results that were obtained for Africa and the Benguela upwelling system. This contribution is particularly interesting as the authors attempt to decipher amongst several mechanisms that can lead to sea-surface temperature changes in the EBUs regions, namely changes in Ekman pumping, changes in surface turbulent fluxes, changes in radiative forcing and horizontal heat advection. Authors show that different mechanism are at play depending if California, South America, or Benguela EBU is considered. The MS will fit well in Climate of the Past, still I suggest some clarifications / improvements that are somewhere

between minor and major.

First, the "uplift history" part could be improved. Despite uncertainties, numerous papers have been published in the last decade that help constraining the elevation history of the different mountain ranges considered. For example: For the Andes, (Garzione et al., 2008, 2014; Leier et al., 2013) . For Africa see (Moucha and Forte, 2011; Wichura et al., 2010, 2015). Having a more complete review of the literature on these paleoelevations could in turn fuel a discussion on the relevance of sensitivity experiments to assess the EBU evolution: If topography was already partly uplifted during the Miocene, would the later phases of uplift involve changes in elevation strong enough to trigger the atmospheric and oceanic dynamics mechanisms invoke in the paper ?

Second, I acknowledge the effort to validate the model, but this part (5.1) is the weakest of the manuscript in its present form. The authors use their control experiment, which they acknowledge have different boundary conditions than present-day (orbital parameters and lad surface conditions, specifically), to compare to data or higher resolution modelling. Moreover they do not provide actual figure of differences of Ekman pumping between their simulation and data/validated model. I would suggest to rewrite this part, use a "true" preindustrial simulation, and compare and show the anomalies with available upwelling climatologies. See for example Yi et al. (Yi et al., 2018) for such climatologies. Lastly, figures show strong Ekman pumping on oceans western boundaries. It would be relevant to explain these signals. I think that at some point, either in part 5.1 or in the discussion, the authors need to discuss the need (or not) of high spatial resolution to correctly represent upwellings in GCMs.

By the way, fig. 4 to fig 6. It would be easier to follow the text if the figures depicted NOTOPO and CTL-NOTOPO, rather than CTL and CTL-NOTOPO.

The results are well-presented, but could be improved by a deeper analysis of the links between uplift and atmospheric physics/dynamics. Some diagnoses (maybe different geopotential heights, slp and air-temperature) could help the reader understand how

surface winds and cloud covers are affected by the topography. I was also wondering if removing the topography would alter subgrid-scale parameterizations of moutain drags, and in turn alter the atmospheric dynamics. The ms would be more complete if authors could elaborate a bit on that.

The cloud radiative forcing (CRF) change between experiments with and without uplifted mountain ranges is well-described and seducing. I think the discussion could still be improved by (1) giving a bit more information about the main characteristics of cloud parameterizations in CCSM3 and (2) mapping the CRF changes both in LW and SW, to confirm the invoked mechanisms. At some point a discussion on CCSM3 ability to represent correctly cloud cover along mountain ranges will be necessary.

Suggested refs:

Garzione, C. N., Hoke, G. D., Libarkin, J. C., Withers, S., MacFadden, B., Eiler, J., Ghosh, P. and Mulch, A.: Rise of the Andes, Science, 320(5881), 1304–1307, doi:10.1126/science.1148615, 2008. Garzione, C. N., Auerbach, D. J., Jin-Sook Smith, J., Rosario, J. J., Passey, B. H., Jordan, T. E. and Eiler, J. M.: Clumped isotope evidence for diachronous surface cooling of the Altiplano and pulsed surface uplift of the Central Andes, Earth Planet. Sci. Lett., 393, 173–181, doi:10.1016/j.epsl.2014.02.029, 2014. Leier, A., McQuarrie, N., Garzione, C. and Eiler, J.: Stable isotope evidence for multiple pulses of rapid surface uplift in the Central Andes, Bolivia, Earth Planet. Sci. Lett., 371–372, 49–58, doi:10.1016/j.epsl.2013.04.025, 2013. Moucha, R. and Forte, A. M.: Changes in African topography driven by mantle convection, Nat. Geosci., 4(10), 707–712, doi:10.1038/ngeo1235, 2011. Wichura, H., Bousquet, R., Oberhänsli, R., Strecker, M. R. and Trauth, M. H.: Evidence for middle Miocene uplift of the East African Plateau, Geology, 38(6), 543–546, doi:10.1130/G31022.1, 2010. Wichura, H., Jacobs, L. L., Lin, A., Polcyn, M. J., Manthi, F. K., Winkler, D. A., Strecker, M. R. and Clemens, M.: A 17-My-old whale constrains onset of uplift and climate change in east Africa, Proc. Natl. Acad. Sci., 112(13), 3910–3915, doi:10.1073/pnas.1421502112, 2015. Yi, X., Hünicke, B., Tim, N. and Zorita, E.: The relationship between Arabian

Sea upwelling and Indian Monsoon revisited in a high resolution ocean simulation, Clim. Dyn., 50(1), 201–213, doi:10.1007/s00382-017-3599-8, 2018.

---

## Author Comment (AC1) · 20 Aug 2019

Response to Reviewer #1 for the Manuscript: "The effect of mountain uplift on eastern boundary currents and upwelling systems" by Gerlinde Jung, Matthias Prange

We are grateful for the referee's additional comments which helped us to further improve the quality of the manuscript.

Anonymous Referee #1 ()

Modern coastal upwelling systems initiated and intensified since the Neogene. However, the reasons for their strengthening throughout the Miocene and Pliocene remain unclear. In the paper, the authors carry out sensitive experiments to investigate the impacts of mountain uplift on the three upwelling systems. The authors carefully diagnose

the model outputs, in particular clearly illustrate the feedbacks behind the upwelling responses. The paper is well written. I would recommend its publication after considering the suggestions below.

General comments: 1. The author should introduce the vertical mixing schemes in the model. In addition to the background vertical mixing, does the model include other vertical mixing param- eterizations, for example the tidal mixing, the eddy mixing. Some of these mixings are also influenced by changes in winds. In other word, when the topography is modified, the changes in winds will also influence these vertical mixings. If these vertical mixings remain unchanged, there are some uncertainties included in the current simulations.

=> We added some information on the vertical mixing scheme of CCSM3 to the model description of paragraph 4.1. Through the KPP scheme wind-stress directly affects vertical mixing coefficients.

2. The uplifts of the Andes and North American Cordillera induce significant cooling around the adjacent upwelling regions. The authors should potentially compare some model outputs with existed proxy data?

=> It is not possible to directly compare our model results to proxy records quantitatively due to the fact that the model experiments do not intend to represent a specific time span in Earth's history and only test the effect of a change in only one boundary condition out of many changes that occurred since the late Miocene. Nevertheless we now confront our model results with the range of SST changes from proxy studies of the major EBUS regions in the discussion section where we also discuss the limitations of our modelling approach. And we added one sentence in the introduction to make the goal of our sensitivity experiments clearer.

3. For the cross-section analysis, I recommend the authors could also do that with an averaged latitude zone over the upwelling regions, especially for the vertical velocity response, rather than using a specific latitude.

=> Since the upwelling area is very limited in its longitudinal extent and additionally the longitudinal position varies largely with latitude it is not feasible to average the signal over latitudinal zones. This would lead to an unnecessary smoothing of the signal. In our opinion it is therefore more useful to investigate the signal at the location of its maximum effect.

4. I am interested in the thermocline depths changes around the three upwelling regions and their potential impacts on the cooling strength.

=> We revised the vertical cross-section plots (Figs. 10-13) by changing the color scales to see the temperature variation with depth in more detail and by adding the temperatures for the control run with high mountain elevation. From visual inspection of these figures it appears that the changes in thermocline depth and structure are quite complex in some cases. As expected, the most evident relationship between surface/upper-ocean cooling and thermocline shoaling is found in the Benguela upwelling region.

Specific comments: 1. Page 3 line 15 : "Neogene" not "Neogen"

=> done

2. Figure 1 and 9-12, Please denote each panel with alphabet letters.

=> done

3. Figure 8b, please explain why choose the depth of 47m here rather than 70m?

=> We chose the depth of 47m in the case of the North American uplift, since this is where the maximum signal of vertical velocity change is found. This is also explained in the manuscript. We now additionally added a note on that in the figure caption (now Figure 9).

---

## Author Comment (AC2) · 20 Aug 2019

Response to Reviewer #2 for the Manuscript: "The effect of mountain uplift on eastern boundary currents and upwelling systems" by Gerlinde Jung, Matthias Prange

We are grateful for the referee's additional comments which helped us to further improve the quality of the manuscript.

Anonymous Referee #2

This manuscript explores the impact of different mountain uplifts on eastern boundary upwelling systems, through a set of sensitivity experiments to topography run with CCSM3 model. It echoes a previous publication by the authors, but this particular ms appears as a generalization assessment of the previous results that were obtained

for Africa and the Benguela upwelling system. This contribution is particularly interesting as the authors attempt to decipher amongst several mechanisms that can lead to sea-surface temperature changes in the EBUs regions, namely changes in Ekman pumping, changes in surface turbulent fluxes, changes in radiative forcing and horizontal heat advection. Authors show that different mechanism are at play depending if California, South America, or Benguela EBU is considered. The MS will fit well in Climate of the Past, still I suggest some clarifications / improvements that are somewhere between minor and major.

First, the "uplift history" part could be improved. Despite uncertainties, numerous papers have been published in the last decade that help constraining the elevation history of the different mountain ranges considered. For example: For the Andes, (Garzione et al., 2008, 2014; Leier et al., 2013) . For Africa see (Moucha and Forte, 2011; Wichura et al., 2010, 2015).

=> We extended the literature review of the uplift histories according to the reviewers' advice.

Having a more complete review of the literature on these paleoel- evations could in turn fuel a discussion on the relevance of sensitivity experiments to assess the EBU evolution: If topography was already partly uplifted during the Miocene, would the later phases of uplift involve changes in elevation strong enough to trigger the atmospheric and oceanic dynamics mechanisms invoke in the paper ?

=> The atmospheric and oceanic changes described in the manuscript refer to elevation changes from 50% to 100%. Hence our sensitivity experiments already showed the changes from an already partly uplifted situation. Other literature indeed showed also effects of an uplift from a completely flat earth (e.g. Feng and Poulsen, 2014) or continent (e.g. Sepulchre et al, 2009 for Southern America). There is definitely a much larger effect in these cases (e.g. completely different air flow over Southern America with flat terrain and already significant blocking of the Westerlies with 50% altitude of

the Andes.

Second, I acknowledge the effort to validate the model, but this part (5.1) is the weakest of the manuscript in its present form. The authors use their control experiment, which they acknowledge have different boundary conditions than present-day (orbital parameters and lad surface conditions, specifically), to compare to data or higher resolution modelling. Moreover they do not provide actual figure of differences of Ekman pumping between their simulation and data/validated model. I would suggest to rewrite this part, use a “true” preindustrial simulation, and compare and show the anomalies with available upwelling climatologies.

=> We changed figures 2-3 (->Fig 3-4) and added a similar figure for the Benguela upwelling region (Fig 2) where we now use the data from a true preindustrial control run and compare this run qualitatively with the data available and described in paragraph 5.1. We have to make the reviewer aware of the fact that this comparison is also hampered by the fact that the respective observational/model data is mostly present-day and not preindustrial.

See for example Yi et al. (Yi et al., 2018) for such climatologies.

=> We could unfortunately not find any upwelling climatologies in the named reference, but we considered some of the cited literature of regional model simulations and added those to the evaluation our model results. Moreover, we post-processed and plotted vertical velocities from the Carton-Giese SODA 2.2.4 reanalysis data for additional comparison in the Supplement.

Lastly, figures show strong Ekman pumping on oceans western boundaries. It would be relevant to explain these signals.

=> We added some information on other upwelling systems (like the Southern Caribbean upwelling) to paragraph 5.1, but did not go into detail, since our publication is concentrating on the Eastern Boundary upwelling systems.

I think that at some point, either in part 5.1 or in the discussion, the authors need to discuss the need (or not) of high spatial resolution to correctly represent upwellings in GCMs.

=> We added some information on higher resolution runs and the benefits of higher resolutions to paragraph 5.1.

By the way, fig. 4 to fig 6. It would be easier to follow the text if the figures depicted NOTOPO and CTL-NOTOPO, rather than CTL and CTL-NOTOPO.

=> We thank the reviewer for making us aware of this inconsistency. We now changed all figures to the scheme CTL and CTL-NOTOPO (hence now all figures are consistant in that). We therefore did not change figures 4-6 (now Fig. 5-7), but added the illustrations for CTRL to figures 10-13.

The results are well-presented, but could be improved by a deeper analysis of the links between uplift and atmospheric physics/dynamics. Some diagnoses (maybe different geopotential heights, slp and air-temperature) could help the reader understand how surface winds and cloud covers are affected by the topography.

=> A deeper analysis of the mechanisms is beyond the scope of this manuscript and might in future publication be discussed for the different upwelling areas. We added some additional information from literature on publications that focused on similar effects to the discussion section.

I was also wondering if removing the topography would alter subgrid-scale parameterizations of moutain drags, and in turn alter the atmospheric dynamics. The ms would be more complete if authors could elaborate a bit on that.

=> Since we do not remove the topography entirely, but only reduce the altitude to 50% of the present-day topography and we leave the sub-grid scale orographic standard deviation untouched (also no paramterizations e.g. gravity-wave drag scheme are switched off), there is no direct effect on sub-grid scale parameterization, but only the

effect through interaction with the changed large-scale flow.

The cloud radiative forcing (CRF) change between experiments with and without uplifted mountain ranges is well-described and seducing. I think the discussion could still be improved by (1) giving a bit more information about the main characteristics of cloud parameterizations in CCSM3 and

=> we added information on the cloud parameterization in CAM3 to the model description

(2) mapping the CRF changes both in LW and SW, to confirm the invoked mechanisms.

=> We added some figures that show the changes in cloud coverage at different levels and the changes in shortwave, as well as long-wave cloud forcing to the Supplement and discussed that issue in paragraph 5.2.4 for the different upwelling regions.

At some point a discussion on CCSM3 ability to represent correctly cloud cover along mountain ranges will be necessary.

=> We added some information on the ability of CCSM3 to simulate low clouds in the discussion.